# How Far Are LLMs from Professional Poker Players? Revisiting Game-Theoretic Reasoning with Agentic Tool Use

**Minhua Lin**[1]    **Enyan Dai**[2]    **Hui Liu**[3]    **Xianfeng Tang**[3]    **Yuliang Yan**[2]    **Zhenwei Dai**[3]
**Jingying Zeng**[3]    **Zhiwei Zhang**[1]    **Fali Wang**[1]    **Hongcheng Gao**[4]    **Chen Luo**[2]
**Xiang Zhang**[1]    **Qi He**[5]    **Suhang Wang**[1]
[1] The Pennsylvania State University    [2] HKUST (GZ)    [3] Amazon
[4] Tsinghua University    [5] Microsoft

## Abstract

As Large Language Models (LLMs) are increasingly applied in high-stakes domains, their ability to reason strategically under uncertainty becomes critical. Poker provides a rigorous testbed, requiring not only strong actions but also principled, game-theoretic reasoning. In this paper, we conduct a systematic study of LLMs in multiple realistic poker tasks, evaluating both gameplay outcomes and reasoning traces. Our analysis reveals LLMs fail to compete against traditional algorithms and identifies three recurring flaws: reliance on heuristics, factual misunderstandings, and a "knowing–doing" gap where actions diverge from reasoning. An initial attempt with behavior cloning and step-level reinforcement learning improves reasoning style but remains insufficient for accurate game-theoretic play. Motivated by these limitations, we propose ToolPoker, a tool-integrated reasoning framework that combines external solvers for GTO-consistent actions with more precise professional-style explanations. Experiments demonstrate that ToolPoker achieves state-of-the-art gameplay while producing reasoning traces that closely reflect game-theoretic principles.

## 1    Introduction

Large Language Models (LLMs) are increasingly deployed in high-stakes domains such as cybersecurity (Ameri et al., 2021) and strategic decision-making (Jiang et al., 2023), where success requires not only factual recall but also reasoning under uncertainty and informed decision-making. A natural testbed for these abilities is *game-playing*, which combines reasoning, planning, and opponent modeling. Poker is especially suitable as a canonical incomplete-information game (Harsanyi, 1995), requiring players to act with hidden information, estimate opponents' ranges, and anticipate future outcomes. Importantly, professional players succeed not only by choosing strong actions, but by *reasoning in a game-theoretic manner* (Brown & Sandholm, 2019), grounding decisions in equilibrium principles while adapting to opponents. Thus, to play like professionals, one must not only act optimally but also *think strategically*. Evaluating LLMs in poker requires going beyond win rate and examining whether their *reasoning traces* reflect principled strategic thinking.

Motivated by this, we ask: *How far are LLMs from professional poker players?* Several recent studies have explored LLMs in such game-theoretic games. For instance, GTBench (Duan et al., 2024) and PokerBench (Zhuang et al., 2025) focus on gameplay outcomes and show that LLMs struggle to compete. Suspicion-Agent (Guo et al., 2023) uses theory-of-mind prompting in Leduc Hold'em, with GPT-4 surpassing neural baselines such as NFSP (Heinrich & Silver, 2016), but still falls short of equilibrium-based methods like CFR+ (Zinkevich et al., 2007). GameBot (Lin et al., 2025c) examines reasoning steps but only measures correctness. While insightful, these works focus narrowly on outcomes, offering limited understandings of *why* LLMs succeed or fail.

To fill this gap, we conduct a systematic study of LLMs in poker, analyzing both gameplay and reasoning traces. Our analysis shows that LLMs consistently underperform traditional baselines, such as NFSP (Heinrich & Silver, 2016) and CFR+ Tammelin (2014), ranging from reinforcement learning (RL) to equilibrium-based solvers, due to three key reasoning flaws: (i) *Heuristic reasoning*: LLMs often rely on shallow heuristics rather than rigorous game-theoretic principles. (ii)

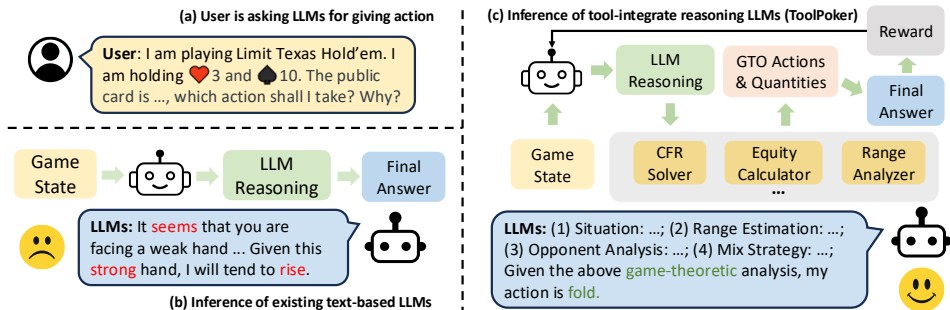

Figure 1: Illustration of ToolPoker and its advantages over LLMs using internal policies.

*Factual misunderstanding*: LLMs sometimes misjudge fundamental aspects of the game, such as hand strength, pot odds, or opponent range estimation, leading to systematically flawed reasoning and (iii) *Knowing–doing gap*: even when LLMs articulate sound reasoning, their final actions often deviate from it, exposing a gap between knowledge expression and decision execution.

To investigate whether these flaws can be mitigated internally, we attempt a two-stage framework: (i) behavior cloning (BC) on expert reasoning traces to instill game-theoretic principles, and (ii) RL fine-tuning with step-level rewards. While this improves fluency and expert-like reasoning style, it remains insufficient for precise derivations or competitive gameplay, underscoring LLMs' fundamental limitations in game-theoretic tasks.

Motivated by these limitations, we pursue an alternative direction: leveraging LLMs' strength in *tool use*. However, achieving this integration in poker is non-trivial and challenging: (i) *Multi-tool dependency*. Accurate game-theoretic reasoning often requires multiple solvers (e.g., action and equity solvers), and naively teaching LLMs to invoke these tools across multi-turn poker scenarios leads to error propagation and unstable training. (ii) *High data cost*. Collecting large-scale reasoning traces augmented with solver calls requires expensive LLM annotation and careful domain-specific tool invocation, making it prohibitively costly to build.

To address these challenges, we introduce **ToolPoker**, the first tool-integrated reasoning (TIR) framework for *imperfect-information games* (Fig. 1), which teaches LLMs to call external poker solvers to provide game-theoretic optimal (GTO) actions and supporting quantities such as equity and hand ranges for accurate expert-level explanations. (i) We design a *unified tool interface* that consolidates solver functionalities into a single API, returning all quantities in one query to simplify tool use and stabilize training. (ii) We construct a *small-scale expert-level* reasoning dataset (Sec. 4.1) inspired by the thought process of professional players, and programmatically augment it with standardized tool invocation templates and execution outputs, ensuring high-quality and reducing annotation cost. This also provides a robust foundation for the following RL training in TIR. By combining GTO-guaranteed computation with human-like reasoning, ToolPoker overcomes fundamental weaknesses of policy-only training and moves LLMs closer to professional-level play. Experiments across multiple poker tasks demonstrate that ToolPoker achieves both state-of-the-art gameplay performance and produces reasoning traces that align much more closely with game-theoretic principles.

Our **main contributions** are summarized as follows: **(i)** We conduct the first systematic study of LLMs in poker, revealing fundamental reasoning flaws such as *heuristic bias*, *factual misunderstanding*, and *knowing–doing gaps*. **(ii)** We make an initial attempt to improve LLMs' internal policies through a two-stage RL framework. While effective at improving reasoning style, this approach remains insufficient for GTO reasoning and accurate game-theoretic derivation. **(iii)** We introduce ToolPoker, a tool-integrated reasoning framework that leverages external solvers to guarantee GTO-consistent actions while enabling LLMs to generate precise, professional-style explanations. **(iv)** Extensive experiments show that ToolPoker achieves state-of-the-art gameplay performance and produces reasoning traces that align closely with professional game-theoretic principles.

## 2 BACKGROUNDS AND PRELIMINARIES

**Two-Player Imperfect Information Poker Games**. In this paper, we explore using LLMs to play poker with imperfect information. Following prior work (Guo et al., 2023; Huang et al., 2024),

we focus on three widely studied two-player variants of increasing complexity: Kuhn Poker, Leduc Hold'em, and Limit Texas Hold'em, where their backgrounds and rules are in Appendix B.

**Game-theoretic Reasoning**. In poker, professional players go beyond heuristics or pattern recognition by systematically evaluating equity, ranges, and pot odds within a game-theoretic framework, guiding them toward actions that converge to Nash equilibrium. An example of such professional-style reasoning is in Appendix B.6, with further details on Nash equilibrium in Appendix B.5.

**Problem Statement.** We model a two-player poker game as a partially observable Markov decision process (POMDP) $(\mathcal{S}, \mathcal{A}, \mathcal{T}, \mathcal{R}, \Omega, O)$, where $\mathcal{S} = \{s^t : 1 \leq t \leq T\}$ is the set of true states, $T$ is the maximum turns, $\mathcal{A}$ is the action space, $\mathcal{T}$ is the transition function, $\mathcal{R}$ is the reward function, $\Omega$ denotes the observation space, and $O$ represents the observation function. At time $t$, the state is $s^t = \{s_{pub}^t, s_{pri(i)}^t, s_{pri(\neg i)}^t\}$, where $s_{pub}^t$ denotes public information (e.g., community cards, betting), and $s_{pri(i)}^t$ and $s_{pri(\neg i)}^t$ are the private cards of player $i$ and the opponent, respectively. Each player $i$ partially observes $o_i^t = (s_{pub}^t, s_{pri(i)}^t) \in \Omega$ and conditions on its history $h_i^t = (o_i^1, a_i^1, \ldots, o_i^t)$ to choose an action $a_i^t \sim \mu_\theta^i(\cdot \mid f(h_i^t))$, where $f$ is a prompt template that converts game states into natural language task descriptions. A full trajectory is $\tau = (s^1, a_1^1, a_2^1, r_1^1, r_2^1, \ldots, s^T, a_1^T, a_2^T, r_1^T, r_2^T)$. The objective for player $i$ is to learn a policy $\mu_\theta^i$ that maximizes the cumulative reward $\sum_{t=1}^T r_i^t$ in the game.

# 3 ARE LLMS GOOD AT POKER? A PRELIMINARY ANALYSIS

In this section, to understand the capabilities of LLMs in playing poker games, we conduct a preliminary analysis to provide initial evidence regarding the strengths and weaknesses of LLMs compared to traditional algorithms for imperfect-information games.

## 3.1 EXPERIMENTAL SETUP

**Tasks**. To quantitatively evaluate the performance of LLMs in poker, we consider two widely studied and popular poker games, Leduc Hold'em and Limit Texas Hold'em (Brown et al., 2019; Steinberger, 2019; Guo et al., 2023), both implemented in the RLCard environment (Zha et al., 2021a).

**Comparison Methods**. Following (Guo et al., 2023), we consider four traditional baselines for imperfect information games: NFSP (Heinrich & Silver, 2016), DQN (Mnih et al., 2015), DMC (Zha et al., 2021b), and CFR+ (Tammelin, 2014). NFSP and DMC are self-play RL methods tailored to imperfect information games, while CFR+ provides a game-theoretic guarantee of convergence to the Nash equilibrium. For the more complex Limit Texas Hold'em environment, where CFR+ is computationally prohibitive, we instead adopt DeepCFR (Brown et al., 2019), a scalable neural extension of CFR+. These baselines cover diverse strategic paradigms, allowing us to assess LLMs against a broad range of opponent types. Details are provided in Appendix C.1.

**Evaluation Protocol**. To ensure the robustness of our evaluation metrics, in our experiment, we run a series of 50 games with fixed random seeds and fixed player positions We then rerun the 50 games with the same fixed random seeds but switched positions for the compared methods. To evaluate the gameplay performance in poker games, we choose the earned chips as the evaluation metric. Specifically, for each individual poker game, each player starts with 100 chips, the small blind is 1 chip, and the big blind is 2 chips.

## 3.2 COMPARISON WITH TRADITIONAL METHOD

**Setting.** We evaluate a suite of representative LLMs spanning a wide range of parameter scales, including Qwen2.5-3B, Qwen2.5-7B, Qwen2.5-72B (Qwen, 2024), Qwen3-8B (Yang et al., 2025), Llama3-8B (Grattafiori et al., 2024), GPT-4.1-mini (OpenAI, 2025), GPT-4o Hurst et al. (2024), and o4-mini (OpenAI, 2024), where the instruction-following versions of these open-source models are adopted. These models are evaluated against the aforementioned traditional baselines.

**Results Analysis.** Table 1 reports the average chip gain of different LLMs against traditional methods in both Leduc Hold'em and Limit Texas Hold'em. From the table, we observe that (i) Most vanilla LLMs, particularly open-source models with smaller scales, underperform relative to tra-

Table 1: Comparison of various vanilla LLMs against different traditional algorithms trained in Leduc Hold'em and Limit Texas Hold'em environments. Each method plays 100 games with varying random seeds and alternated player positions. Results report net chip gains. In Leduc Hold'em, values range from 1 to 14 chips; in Limit Texas Hold'em, they range from 1 to 99 chips. **Bold** and underline indicate the best and worst performance in each column, respectively. The "Avg." columns summarize LLMs' mean performance across the four traditional baselines.

| | Leduc Hold'em | | | | | Limit Texas Hold'em | | | | |
|---|---|---|---|---|---|---|---|---|---|---|
| | NFSP | DQN | DMC | CFR+ | Avg. | NFSP | DQN | DMC | DeepCFR | Avg. |
| Qwen2.5-3B | -143.5 | -161 | -124 | -114 | -135.5 | -131.5 | -232.5 | -136 | -323.5 | -205.8 |
| Qwen2.5-7B | -57.5 | -93 | -73 | -68.5 | -73.0 | -53.5 | -188 | -144 | -101.0 | -121.6 |
| Qwen2.5-72B | +24.5 | -18 | **-18** | -25 | -9.1 | +6 | -53.5 | -73.5 | **-57.5** | -44.6 |
| Qwen3-8B | -72 | -75 | -75 | -54 | -69.0 | -69 | -73.5 | -69 | -73.5 | -71.2 |
| LLama3-8B | -77.5 | -108.5 | -90 | -71 | -86.7 | +8 | -177.5 | -58 | -206.5 | -108.5 |
| GPT-4.1-mini | **+41.5** | **+60.5** | -22 | -24 | **+14.0** | **+43** | +0 | **-24** | -205.0 | -46.7 |
| GPT-4o | +34 | +53 | -43 | **-8** | +9.0 | -40.5 | -45.5 | -32 | -167.0 | -71.2 |
| o4-mini | +11 | +20 | -33.5 | **-8** | -2.6 | -105 | **+111** | -58 | -117.0 | **-42.2** |

ditional methods. This highlights the limited effectiveness of state-of-the-art LLMs in poker. (ii) CFR+ consistently outperforms all LLMs, including strong closed-source models such as GPT-4o and o4-mini. This is expected, as CFR+ explicitly targets Nash equilibrium strategies, underscoring the importance of game-theoretic reasoning in imperfect-information games. (iii) Against non-equilibrium baselines (i.e., NFSP, DQN, DMC), some large-scale and closed-source LLMs demonstrate competitive or superior performance. For instance, GPT-4o achieves $+41.5$, $+60.5$, and $-22$ chip outcomes against NFSP, DQN, and DMC, respectively. In contrast, small open-source LLMs (e.g., Qwen2.5-3B) exhibit severe losses across all baselines (e.g., $-143.5$, $-161$, and $-124$ chips). These results suggest that while LLMs cannot approximate Nash equilibrium strategies, sufficiently large models can exploit non-equilibrium opponents.

## 3.3 IN-DEPTH ANALYSIS: DECOMPOSING REASONING FLAWS OF LLMS

To understand why LLMs fail to compete with traditional methods in poker, we conduct an in-depth analysis of their reasoning processes. Specifically, we first present several case studies that highlight three key flaws in LLM reasoning, followed by a quantitative analysis to further validate and interpret these observations.

**Case Study of LLMs' Reasoning Flaw**. To probe LLMs' decision-making, we examine their reasoning traces in specific scenarios against baseline opponents. Representative cases from Qwen2.5-3B and GPT-4o are shown in Table 13 and 14 in Appendix C.2. From these examples, we identify three recurrent flaws: (i) *Heuristic Reasoning*. LLMs frequently rely on heuristic-driven reasoning, making decisions based on surface-level patterns or intuitive analogies rather than on rigorous game-theoretic principles. In contrast, the Nash-equilibrium algorithm CFR+ consistently achieves the strongest performance, underscoring the value of game-theoretic reasoning in imperfect-information games like poker. The absence of such equilibrium-oriented reasoning substantially constrains the gameplay performance of LLMs. These two findings indicate that while LLMs are capable of articulating plausible strategic reasoning, their actual decision-making remains constrained by executional inconsistencies and heuristic biases. These limitations ultimately hinder their effectiveness in complex poker games that require advanced strategic reasoning capabilities. (ii) *Factual Misunderstanding*. LLMs often ground their reasoning in intuitive analogies, making them prone to misjudging fundamental aspects of the game, such as hand strength or opponent range estimation. These factual inaccuracies can cascade into flawed reasoning chains and ultimately suboptimal actions. For example, as shown in Tab. 14, GPT-4o incorrectly judged (♠K, ♣10) as weak and preferred folding. However, an equity calculator shows this hand has about $60\%$ equity, indicating it is relatively strong. (iii) *Knowing–Doing Gap*. LLMs often exhibit a mismatch between articulated reasoning and final actions. For instance, in Tab. 13, Qwen2.5-3B correctly reasons that (♣3 ♡10) is not a strong hand and fold is optimal, while it yet proceeds to raise. Such inconsistencies reveal a breakdown between reasoning and execution. Additional case studies are provided in Appendix C.2.

**Quantitative Analysis of LLMs' Reasoning Flaws**. To validate the reasoning flaws observed in case studies, we adopt the LLM-as-a-Judge framework (Dubois et al., 2023). We design three metrics: heuristic reasoning (HR), factual alignment (FA), and action–reasoning consistency (AC), and score each reasoning trace on a 0–2 scale using GPT-4.1-mini as the judge. Metric definitions, judge prompts, and human–LLM agreement are in Appendix C.3 and C.5. For each model, we sample 20

traces and evaluate Qwen2.5-3B/7B/72B, GPT-4.1-mini, and o4-mini. To ensure reliability of LLM-based judging, we manually curate 20 professional-style reasoning traces and score them by LLMs. We observe high agreement with human judgement and include it as a reference (see Appendix C.5).

We report results in Tab. 2. Three key findings are observed: (i) *Reasoning flaws persist across all models.* Qwen2.5-3B scores only 0.53 HR, 0.18 FA, and 1.53 AC, while o4-mini, the strongest model, reaches 1.80/1.56/1.85, still below perfect consistency. This shows systemic heuristic, factual, and knowing–doing flaws in LLMs. (ii) *Scaling improves but does not eliminate flaws.* Larger models (Qwen2.5-72B, o4-mini) improve all metrics, but significant FA and AC gaps remain, showing scale alone cannot achieve professional-level reasoning. (iii) *Action–reasoning consistency remains imperfect.* AC stabilizes around 1.53–1.87, below the professional baseline of 2.0, with o4-mini still exhibiting knowing–doing mismatches. Full details are in Appendix C.4.

Table 2: LLM-as-a-Judge score (0-2) evaluating reasoning traces of various LLMs in Leduc Hold'em and Limit Texas Hold'em. **Bold** and underlined numbers indicate the best and worst performance, respectively.

| | **Leduc Hold'em** | | | | **Limit Texas Hold'em** | | | |
|---|---|---|---|---|---|---|---|---|
| | HR | FA | AC | Avg. | HR | FA | AC | Avg. |
| Professional | 2 | 2 | 2 | 2 | 2 | 2 | 2 | 2 |
| Qwen2.5-3B | 0.53 | 0.18 | 1.53 | 0.74 | 0.55 | 0.30 | 1.60 | 0.81 |
| Qwen2.5-7B | 0.95 | 0.86 | 1.68 | 1.16 | 1.00 | 0.87 | 1.70 | 1.19 |
| Qwen2.5-72B | 1.03 | 1.23 | 1.78 | 1.34 | 1.03 | 1.52 | 1.85 | 1.46 |
| GPT-4.1-mini | 0.98 | **1.73** | **1.87** | 1.52 | 0.95 | 1.61 | 1.87 | 1.47 |
| o4-mini | **1.80** | 1.56 | 1.85 | **1.73** | **1.57** | **1.65** | **1.88** | **1.70** |

Overall, these findings quantitatively reinforce our case studies: despite improvements in scale and instruction tuning, current LLMs remain far from professional-level poker reasoning. They continue to exhibit heuristic biases, factual misunderstandings, and executional inconsistencies that fundamentally limit their game-theoretic reasoning capabilities.

## 4 CAN WE IMPROVE LLMS IN POKER? FAILURES AND INSIGHTS

Building on the preliminary analysis of LLM limitations in poker, we next explore how to improve their ability to both *act* and *reason* like professional players. A natural starting point is supervised fine-tuning (SFT) on expert gameplay. However, while obtaining expert actions is straightforward using established solvers such as CFR+, constructing large-scale datasets with high-quality reasoning traces is extremely costly, making pure SFT impractical at scale. For instance, Wang et al. (2025) report that mastering even simplified poker games like Leduc Hold'em requires at least 400k action-only instances. Adding reasoning traces would multiply both time and financial costs, rendering such datasets infeasible to construct. To address this, inspired by recent progress in RL for enhancing LLM reasoning (Guo et al., 2025) and by traditional RL for poker (Heinrich & Silver, 2016), we make an initial attempt to propose a two-stage framework, BC-RIRL, that combines behavior cloning (BC) with regret-inspired policy optimization (RIRL). In the first stage, BC aims to provide a small but valuable foundation of expert play and reasoning. In the second stage, RIRL refines these policies toward GTO play under Nash–equilibrium–based supervision.

### 4.1 BEHAVIOR CLONING

We first leverage BC to expose LLMs to professional-style reasoning. Following recent advances in reasoning-augmented datasets (Muennighoff et al., 2025) and inspired by professional players's thought process (Appendix B.6), we curate a dataset of professional-level trajectories $\mathcal{D}_b = \{(h^t, a^t, r^t)\}$, where $h^t$ is the full interaction history up to time $t$ and $a^t$ is the corresponding expert response. Expert actions $a^t$ are obtained by querying the state-of-the-art CFR+ solver (Tammelin, 2014) with $h^t$, ensuring alignment with Nash-equilibrium play. Reasoning traces $r^t$ are generated using an LLM guided by domain-specific prompt templates covering key concepts such as hand equity, pot odds, and opponent ranges, to mimic the explanatory style of professional players. The construction prompts and dataset examples are in Appendix D.3. To ensure dataset quality, we implement an automated pipeline that (i) checks consistency between the annotated actions and CFR+ outputs, and (ii) filters out low-quality samples using our HR/FA/AC metrics. After filtering, we obtain a compact dataset of approximately 5k reasoning-augmented samples, which is then used to fine-tune the LLM policy $\pi_\theta$ via supervised fine-tuning (SFT) to imitate expert responses:

$$\mathcal{L}_{\text{BC}} = -\mathbb{E}_{(h^t, a^t) \sim \mathcal{D}_b}[\log \pi_\theta(a^t | h^t)]. \tag{1}$$

This imitation phase grounds the LLM in domain knowledge and equips it with basic game-theoretic reasoning capability. As shown in Sec. 4.3, BC primarily serves as a warm start, providing a crucial foundation for the subsequent RL stage.

## 4.2 REGRET-INSPIRED RL FINE-TUNING

As an initial attempt to refine policies beyond imitation, we attempt a regret-inspired reinforcement learning (RIRL) framework. To overcome the sparse and noisy outcome-based rewards in multi-turn poker games such as Leduc Hold'em and Texas Hold'em, we experiment with a step-level regret-guided reward that leverages signals from a pre-trained CFR solver to guild LLMs minimize cumulative regret and convergence to the Nash equilibrium. Full details of RIRL are in Appendix D.1.

**Regret-guided Reward Design**. Motivated by CFR's success in poker playing by approaching Nash equilibrium from Sec. 3.2, we optimize LLMs via regret minimization. Our key idea is to compute cumulative regrets from a pre-trained CFR solver and normalize them into fine-grained reward signals that capture each action's relative contribution. For a policy $\pi_\theta$ as player $i$, the reward of action $a_i^t$ is defined as:

$$R(a_i^t) = \frac{R_t(a_i^t) - \text{mean}(\{r_t(a_j)\}_{j=1}^{|\mathcal{A}|})}{F_{\text{norm}}(\{r_t(a_j)\}_{j=1}^{|\mathcal{A}|})}, \tag{2}$$

where $F_{\text{norm}}$ denotes a normalization factor, chosen as the standard deviation in our implementation. $r_t(a_i^t)$ is the cumulative regret of action $a_i^t$, indicating how much better or worse it performs compared to the current mixture strategy across time.

**Fine-tuning Objective**. Based on this signal, we fine-tune LLM policy via PPO (Schulman et al., 2017) with the following clipped RL objective:

$$\mathcal{L}_{\text{PPO}}(\theta) = -\mathbb{E}_{x \sim \mathcal{D}_s, y \sim \pi_{old}(\cdot|x)}$$
$$\left[ \min\left( \frac{\pi_\theta(y|x)}{\pi_{old}(y|x)} A, \text{clip}\left( \frac{\pi_\theta(y|x)}{\pi_{old}(y|x)}, 1 - \epsilon, 1 + \epsilon \right) \right) - \beta \mathbb{D}_{\text{KL}}(\pi_\theta(\cdot|c)||\pi_{ref}(y|x)) \right], \tag{3}$$

where $\pi_\theta$ and $\pi_{old}$ denote the current and previous policy models, respectively. $\epsilon$ is the clipping threshold. $\pi_{ref}$ is the reference policy that regularizes $\pi_\theta$ update via a KL-divergence penalty, measured and weighted by $\mathbb{D}_{KL}$ and $\beta$, respectively. Generalized Advantage Estimation (GAE) (Schulman et al., 2015) is used for advantage estimate $A$. $x$ denotes the input samples drawn from $\mathcal{D}$, which is composed of trajectories generated by the current policy $\pi_\theta$. $y$ is the generated outputs via policy LLMs $\pi_\theta(\cdot|x)$. The trajectory collection procedure is introduced in Appendix D.4.

## 4.3 EXPERIMENT ANALYSIS

**Experimental Setup**. Following the settings in Sec. 3.1, we implement BC-RIRL by fine-tuning LLMs with both BC and RIRL, and compare against traditional algorithms as well as LLM-based approaches. For traditional baselines, we adopt NFSP, DQN, DMC, and CFR+, consistent with Sec. 3.1. For LLM-based baselines, in addition to direct prompting without fine-tuning, we consider two variants: (i) **BC-SPRL**, which fine-tunes LLMs through BC and self-play RL with sparse outcome-based rewards, and (ii) **RIRL**, which fine-tunes LLMs with RIRL alone, without the BC stage. Further details of SPRL are in Appendix E. Other settings follow these in Sec. 3.1, including the evaluation metrics. The implementation details are in Appendix D.5.

**Comparison Results**. We fine-tune Qwen2.5-7B with BC-RIRL and compare against traditional algorithms and vanilla LLMs. The gameplay and reasoning results are reported in Tab. 3 and Tab. 4.

*Gameplay*. **(i)** All RL-based fine-tuning variants improve performance in Kuhn Poker, showing that both outcome- and regret-based feedback provide useful signals in simple environments. **(ii)** BC-RIRL outperforms direct prompting and BC-SPRL (e.g., +17.0 chips vs. GPT-4.1-mini) but still trails CFR+ (−34.0 chips) In Leduc Hold'em, indicating dense regret feedback is more effective than sparse outcome rewards in complex poker games, yet **insufficient** for equilibrium-level play. **(iii)** Pure RIRL without the BC stage does not yield improvements in Leduc Hold'em (−64.5 chips vs. GPT-4.1-mini), highlighting BC as a necessary foundation.

*Reasoning*. (i) RIRL consistently improves HR and AC (e.g., 1.93 HR and 1.90 AC in Leduc Hold'em vs. 1.80/1.85 for o4-mini), reducing heuristic flaws and the knowing–doing gap. (ii) RIRL

Table 3: Results of comparison fine-tuning methods against various traditional-based and vanilla LLMs in Kuhn and Leduc Hold'em environment. Other settings follow these in Tab. 1. **Bold** and underlined numbers indicate the best and worst performance, respectively.

| Method | Traditional Methods | | | | Vanilla LLMs | | | | |
| --- | --- | --- | --- | --- | --- | --- | --- | --- | --- |
| | NFSP | DQN | DMC | CFR+ | Qwen2.5-3B | Qwen2.5-7B | GPT-4.1-mini | o4-mini | Avg. |
| *Kuhn* | | | | | | | | | |
| Qwen2.5-7B | -22.0 | -53.0 | -33.0 | -36.0 | +26 | - | -41 | -43 | **-28.8** |
| Qwen2.5-7B$_{RIRL}$ | -14.0 | +3.0 | +10.0 | -5.0 | +43.0 | +8.0 | -1.0 | -11.0 | +4.1 |
| Qwen2.5-7B$_{BC-SPRL}$ | +6.0 | -6.0 | +13.0 | -14.0 | +32.0 | +23.0 | +22.0 | +10.0 | +10.7 |
| Qwen2.5-7B$_{BC-RIRL}$ | +4.0 | +8.0 | +11.0 | -2.0 | +57.0 | +27.0 | +21.0 | +11.0 | **+17.1** |
| *Leduc Hold'em* | | | | | | | | | |
| Qwen2.5-7B | -57.5 | -93.0 | -73.0 | -68.5 | +48.5 | - | -59.5 | -32.5 | -47.9 |
| Qwen2.5-7B$_{RIRL}$ | -42.5 | -80 | -59.5 | -55.0 | +52.0 | +12.0 | +2.5 | -18.5 | -23.6 |
| Qwen2.5-7B$_{BC-SPRL}$ | -93.0 | -154.5 | -95.5 | -103.5 | +2.0 | -18.0 | -64.5 | -54.5 | -72.6 |
| Qwen2.5-7B$_{BC-RIRL}$ | -37.0 | -64.5 | -43.5 | -34.0 | +54.0 | +28.5 | +17.0 | +1.0 | **-9.8** |

Table 4: LLM-as-a-Judge score (0-2) evaluating reasoning traces of various LLMs in two realistic poker tasks. **Bold** and underlined numbers indicate the best and worst performance, respectively.

| | Leduc Hold'em | | | | Limit Texas Hold'em | | | |
| --- | --- | --- | --- | --- | --- | --- | --- | --- |
| | HR | FA | AC | Avg. | HR | FA | AC | Avg. |
| Qwen2.5-7B | 0.95 | 0.86 | 1.68 | 1.16 | 1.00 | 0.87 | 1.70 | 1.19 |
| GPT-4.1-mini | 0.98 | **1.73** | 1.87 | 1.52 | 0.95 | 1.61 | 1.87 | 1.47 |
| o4-mini | 1.80 | 1.56 | 1.85 | **1.73** | 1.57 | **1.65** | 1.88 | **1.70** |
| Qwen2.5-7B$_{RIRL}$ | 0.94 | 0.89 | 1.64 | 1.15 | 0.98 | 0.93 | 1.71 | 1.20 |
| Qwen2.5-7B$_{BC-SPRL}$ | 1.89 | 0.88 | 1.66 | 1.47 | 1.87 | 0.86 | 1.64 | 1.45 |
| Qwen2.5-7B$_{BC-RIRL}$ | **1.93** | 1.06 | 1.86 | 1.61 | **1.88** | 1.12 | 1.87 | 1.59 |

gains only marginal improvement in FA (1.12, 0.87 and 1.65 for RIRL, Qwen2.5-7B and o4-mini), showing that factual misunderstandings remain the main limitation. Together with the case studies, these results indicate that while BC-RIRL improves strategic reasoning and action–reasoning alignment, **factual misunderstandings** remain a notable challenge. Full analysis are in Appendix D.2.

**Takeaway**. Our experiments validate that current LLMs are inherently weak at strategic reasoning in game-theoretic tasks. RL fine-tuning with step-level or outcome-based rewards yields modest gameplay gains but still lags behind traditional methods like CFR. Importantly, while our two-stage approach helps LLMs imitate professional reasoning styles, they continue to struggle with precise derivation such as equity and hand ranges. This reveals a fundamental *limitation*: LLMs alone cannot yet achieve both GTO actions and precise reasoning. To bridge this gap, we next explore augmenting LLMs with *tool use*, leveraging their natural strength in tool invocation to support GTO-consistent actions and precise game-theoretic reasoning.

## 5 ToolPoker: Game-theoretic Reasoning with Agentic Tool Use

Building on our analysis in Sec. 4, which highlights the limitations of LLMs in producing GTO actions and precise game-theoretic reasoning, we propose ToolPoker, a tool-integrated reasoning (TIR) framework to leverage LLMs' strength in *tool use* to empower LLMs to leverage external poker solvers to refine their actions and reasoning qualities, which is shown in Fig. 1. To make this tool usage stable and effective, we introduce a unified tool interface that consolidates multiple poker solvers (e.g., CFR and equity calculators) into a single API to simplify this into a single-turn tool use. On the training side, we adopt a two-stage strategy: first, behavior cloning on a code-augmented dataset to teach the model when and how to call external tools; and second, reinforcement learning with a composite reward to further optimize solver integration and reasoning quality.

### 5.1 Tool-Integrated Game-theoretic Reasoning in Poker

**Rollout Process**. To enable GTO-consistent TIR, we design a structured prompt template in Tab. 21 to guide LLM to leverage external poker solvers for game-theoretic reasoning. Concretely, given a policy LLM $\pi_\theta$ as player $i$ at time $t$, $\pi_\theta$ generates a reasoning trace enclosed in `<think></think>` tags. To obtain GTO actions and other quantities, $\pi_\theta$ issues a query in `<tool></tool>` tags, which calls the unified solver interface and returns results wrapped in `<output></output>` tags. These outputs are then incorporated into the reasoning trace before $\pi_\theta$ produces the final action $a_i^t$ within `<answer></answer>` tags.

**Unified Tool Inference**. Obtaining GTO actions and supporting quantities (e.g., equity, pot odds, and range distributions) often requires multiple tool calls, such as a CFR solver and an equity cal-

Table 5: Comparison of various LLM-based methods against different traditional algorithms trained in Leduc Hold'em and Limit Texas Hold'em environments. Other settings follow these in Tab. 1. **Bold** and underline indicate the best and worst performance in each column, respectively.

| | Leduc Hold'em | | | | | Limit Texas Hold'em | | | | |
|---|---|---|---|---|---|---|---|---|---|---|
| | NFSP | DQN | DMC | CFR+ | Avg. | NFSP | DQN | DMC | DeepCFR | Avg. |
| Qwen2.5-7B | -57.5 | -93 | -73 | -68.5 | -73.0 | -53.5 | -188 | -144 | -101.0 | -121.6 |
| Qwen2.5-72B | +24.5 | -18 | -18 | -25 | -9.1 | +6 | -53.5 | -73.5 | -57.5 | -44.6 |
| o4-mini | +11 | +20 | -33.5 | -8 | -2.6 | -105 | **+111** | -58 | -117.0 | -42.2 |
| Qwen2.5-7B$_{BC\text{-}RIRL}$ | -37.0 | -64.5 | -43.5 | -34.0 | -40.5 | -77.5 | -82.5 | -80.5 | -70.2 | -77.6 |
| Qwen2.5-7B$_{ToolPoker}$ | +11.5 | +18.0 | +1.0 | -3.0 | +6.8 | +60.5 | +63.0 | 61.5 | -5.0 | +45.0 |

(a) Reasoning - Leduc   (b) Reasoning - Limit   (c) Ablation - Gameplay   (d) Ablation - Reasoning

Figure 2: Results for ToolPoker: (a) and (b) present reasoning analysis in Leduc Hold'em and Limit Texas Hold'em; (c) and (d) show ablation studies on gameplay and reasoning in Leduc Hold'em.

culator. To simplify and stabilize training, we unify these functionalities into a single standardized interface that provides both the solver's actions and auxiliary statistics for game-theoretic reasoning.

## 5.2 TRAINING ALGORITHM

**BC for TIR**. To construct high-quality TIR data without incurring prohibitive annotation cost, we build an automated pipeline that programmatically augments the reasoning dataset from Sec. 4.1 with standardized tool invocation templates (e.g., `<tool></tool>`) and execution outputs (e.g., `<output></output>`). This resulting dataset $\mathcal{D}_c$ is then used to train ToolPoker via SFT, providing a foundation for LLMs to know how to invoke tools for game-theoretic reasoning. The realistic example and the details of the automatic pipeline are in Tab. 22 in Appendix G.2.

**RL Fine-tuning**. We train ToolPoker with PPO (Schulman et al., 2017), where the objective function is defined in Eq. (8). To better support TIR, we follow ReTool (Feng et al., 2025a) and integrate external poker solvers into the LLM policy $\pi_\theta$, enabling multi-turn real-time tool use that provides GTO-consistent actions and supporting quantities from external tools. To guide the training, we design a composite reward function. Formally, given player $i$ at time step $t$, the reward is defined as

$$R(a_i^t, \hat{a}_i^t, \rho_i^t) = R_{\text{answer}}(a_i^t, \hat{a}_i^t) + \alpha_f \cdot R_{\text{format}}(\rho_i^t) + \alpha_t \cdot R_{\text{tool}}(\rho_i^t), \tag{4}$$

where $a_i^t$ is the ground-truth action from the CFR solver, $\hat{a}i^t$ is the model-predicted action, and $\rho_i^t$ is the generated reasoning trace. Here, $R_{\text{answer}}$, $R_{\text{format}}$, and $R_{\text{tool}}$ correspond to the answer reward, format reward, and tool-execution reward, respectively, ensuring that ToolPoker not only outputs GTO-consistent actions but also generates structured reasoning traces with effective tool usage. $\alpha_f$ and $\alpha_t$ are the weights to balance the impact of format and tool execution rewards. More details of these reward functions are in Appendix G.3. The fine-tuning algorithm is in Alg. 1 of Appendix G.4.

## 5.3 EXPERIMENTAL RESULTS

**Evaluation Setup**. We conduct evaluations on two realistic and complex poker tasks, Leduc Hold'em and Limit Texas Hold'em. We compare ToolPoker with the following baselines: (i) Traditional algorithms: NFSP, DQN, DMC, and CFR; (ii) Vanilla LLMs: Qwen2.5-7B, Qwen2.5-72B, and o4-mini; (iii) Fine-tuning-based baseline: BC-RIRL. Other settings follow these in Sec. 4.3. More Implementation details of ToolPoker are in Appendix G.5.

**Gameplay Performance**. We first explore the gameplay performance of ToolPoker. Qwen2.5-7B is the base model for fine-tuning. We compare ToolPoker with BC-RIRL and three vanilla LLMs, Qwen2.5-7B, Qwen2.5-72B and o4-mini, where the comparison results are reported in Tab. 5. Two key findings emerge: (i) ToolPoker *achieve state-of-the-art gameplay perfomrance against traditional algorithms*. For instance, ToolPoker gains $+60.5$, $+63.0$ and $+61.5$ chips against NFSP, DQN and DMC in Limit Texas Hold'em, while BC-RIRL gains $-77.5$, $-82.5$ and $-80.5$ chips against them. This indicates the effectiveness of ToolPoker in calling CFR solver to obtain GTO-consistent

actions. (ii) ToolPoker *slightly underperforms CFR but is still comparable in both poker environments*. Specifically, ToolPoker gain $-3.0$ and $-5.0$ chips against CFR+ and DeepCFR in both Leduc Hold'em and Limit Texas Hold'em, which are minor. We analyze the reason is that while ToolPoker provides a high success rate in executing the CFR solver to provide GTO-consistent action, it is inevitable that occasional errors occur in tool calling.

**Reasoning Quality**. To assess whether ToolPoker also improves *reasoning*, we employ the LLM-as-a-Judge framework following the settings in Sec. 4.3. Fig. 2 (a) and (b) summarize the results across three metrics. Two observations emerge: (i) ToolPoker *achieves near-perfect across all three scores*, outperforming all baselines and approaching professional levels. This indicates that, beyond delivering state-of-the-art gameplay performance, ToolPoker also enables LLMs to generate precise and logically consistent reasoning traces grounded in game-theoretic principles. (ii) Compared with BC-RIRL, ToolPoker *yields substantially higher FA scores*. This demonstrates the importance of leveraging external solvers: while BC-RIRL can articulate plausible reasoning, it often lacks accurate auxiliary quantities (e.g., equities, ranges). In contrast, ToolPoker grounds its reasoning in solver-derived calculations, ensuring rigor and internal consistency.

**Ablation Studies.** To understand the impact of each component in ToolPoker, we implement two ablated variants: (i) **ToolPoker/BC**: removes BC and learns tool use only via RL; (ii) **ToolPoker/RL**: discards RL fine-tuning and relies solely on BC. We measure both gameplay performance (against NFSP and DQN) and reasoning quality in Leduc Hold'em, with results shown in Fig. 2 (c) and (d). The full ToolPoker achieves the strongest overall performance, while the variants reveal complementary weaknesses. Specifically: (i) ToolPoker*/BC suffers from lower HR and weaker gameplay*, suggesting it can query the solver but fails to internalize game-theoretic reasoning patterns; (ii) ToolPoker*/RL attains higher HR but performs poorly in gameplay and FA/AC*, indicating it imitates reasoning superficially without aligning with GTO-consistent actions. These results highlight that BC provides the foundation for TIR, while RL fine-tuning aligns solver execution with GTO actions and precise derivation. Together, they enable ToolPoker to learn not only how to call the solver, but also how to integrate outputs into coherent, professional-style reasoning traces. More discussions are in Appendix G.6.

## 6 RELATED WORK

**Strategic Reasoning in LLMs.** Recent studies have examined LLMs in game-theoretic settings, including poker (Duan et al., 2024; Zhai et al., 2024; Zhuang et al., 2025; Wang et al., 2025). Unlike prior work that primarily evaluates gameplay outcomes, we also analyze the *reasoning process*, identifying why LLMs fail to achieve GTO play. Moreover, we introduce the first TIR framework that leverages poker solvers for professional-level gameplay. Further discussion is in Appendix A.1.

**Tool Learning on LLMs.** TIR equips LLMs with external tools for domains such as math and web search (Gao et al., 2023; Jin et al., 2025), which are typically fully observed and single-agent. In contrast, ToolPoker extends TIR to imperfect-information games, integrating poker solvers to ensure GTO actions and rigorous reasoning. Full details on RL and TIR are in Appendix A.2 and A.3.

## 7 CONCLUSIONS AND FUTURE WORKS

In this paper, we revisit strategic reasoning in LLMs through poker with imperfect information. Our analysis shows that current LLMs fall short of professional-level play, exhibiting heuristic biases, factual misunderstandings, and a knowing–doing gap between their reasoning and actions. An initial attempt with BC and RIRL partially reduces heuristic flaws but is still not enough for precise game-theoretic derivations or competitive gameplay. To address this, we introduce **ToolPoker**, a TIR framework that leverages LLMs' strength in tool use to incorporate external poker solvers. ToolPoker enables models not only to call solvers for GTO actions but also to ground their rigorous and accurate game-theoretic reasoning in solver outputs. Experiments across multiple poker tasks show that ToolPoker achieves state-of-the-art gameplay performance and produces reasoning traces that align closely with professional game-theoretic principles. Our research paves the way for further exploration of TIR in more complex strategic settings, shifting the focus beyond solely improving models' internal policies. Further discussion of future works is provided in Appendix I.

## 8 ETHICS STATEMENT

This paper studies LLMs in the context of poker as a rigorous benchmark for strategic reasoning under uncertainty. While poker involves gambling in practice, our experiments are conducted entirely in simulated environments without any financial transactions or human participants. Thus, this research does not pose risks related to gambling addiction or monetary harm.

Our contributions focus on methodology and evaluation. We study the reasoning capabilities of LLMs, propose new training frameworks, and benchmark them against both traditional algorithms and LLM-based methods. These findings aim to deepen understanding of LLM reasoning in imperfect-information games, with potential implications for broader domains such as cybersecurity and negotiation. We acknowledge that advanced poker agents could, if misused, be deployed in real-money contexts. To mitigate this risk, we release code and datasets solely for research purposes, emphasizing their use as benchmarks for safe and reproducible evaluation.

Finally, we ensured that no personally identifiable or sensitive human data were used in this work. All datasets are synthetically generated using poker solvers or LLMs. We believe the potential benefits of this paper, including advancing understanding of the limitations of LLMs' reasoning, improving the design of tool-augmented AI, and supporting safer deployment in high-stakes domains, clearly outweigh the minimal risks.

## 9 REPRODUCIBILITY STATEMENT

We have made every effort to ensure reproducibility. The details of our proposed methods, including model architectures, training objectives, and hyperparameters, are provided in Sec. 4 and Sec. 5. Experimental setups, including datasets, preprocessing steps, and evaluation protocols, are described in Sec. 3.1, Sec. 4.3, and Sec. 5.3, with additional details in the Appendix.

## ACKNOWLEDGMENT

This material is based upon work supported by, or in part by the Army Research Office (ARO) under grant number W911NF-21-10198 and the Department of Homeland Security (DHS) under grant number 17STCIN00001-05-00. The views and conclusions contained in this material are those of the authors and should not be interpreted as necessarily representing the official policies, either expressed or implied, of the funding agencies.

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

# A  FULL DETAILS OF RELATED WORKS

## A.1  STRATEGIC REASONING IN LLMs

With the rapid progress of LLMs' cognitive capabilities, recent studies have begun to investigate their potential for strategic reasoning in game-theoretic settings (Duan et al., 2024; Gupta, 2023; Huang et al., 2024; Zhuang et al., 2025; Wang et al., 2025). GTBench (Duan et al., 2024) introduces a comprehensive benchmark covering a variety of games to assess LLMs' ability to follow equilibrium principles. Gupta (2023) provide one of the first empirical evaluations of GPT-4 and ChatGPT in poker, revealing systematic deviations from GTO gameplay. Guo et al. (2023) explore theory-of-mind (ToM) prompting in Leduc Hold'em, showing that GPT-4 with ToM reasoning can outperform neural baselines such as NFSP (Heinrich & Silver, 2016). PokerGPT (Huang et al., 2024) fine-tunes LLMs on poker-specific data and observes improvements in gameplay, while PokerBench (Zhuang et al., 2025) constructs a benchmark on No-Limit Hold'em. More recently, Wang et al. (2025) curate large-scale action-only datasets (more than 400k+ examples) and demonstrate gains in card games by fine-tuning LLMs on such data. Additional works (Costarelli et al., 2024; Herr et al.) also investigate gameplay performance and biases of LLMs in other strategic games, such as Tic-Tac-Toe and Prisoner's Dilemma. In addition to exploring strategic reasoning in text-based settings, other works study to use RL in the multimodal domain (Lin et al., 2024; Zhai et al., 2024). For instance, Zhai et al. (2024) extend this line of work to the multimodal domain by fine-tuning large vision–language models (VLMs) with RL. This paper leverages CoT-style intermediate reasoning to guide VLMs through multi-step decision-making tasks, including poker. This demonstrates that RL can enable VLMs to effectively explore and execute visual–textual reasoning sequences.

Our work differs in two key aspects: (i) unlike prior works that mainly evaluate or improve LLMs' *actions*, we further analyze their *reasoning process*, asking how LLMs think before acting and why they fail to achieve GTO play; and (ii) rather than relying on internal policies alone, we propose the first tool-integrated reasoning framework that leverages poker solvers, enabling both equilibrium-consistent actions and professional-style game-theoretic reasoning.

## A.2  REINFORCEMENT LEARNING

Reinforcement Learning (RL) has emerged as a powerful mechanism for enhancing the reasoning abilities of LLMs. In context of LLMs, RL was first introduced through Reinforcement Learning from Human Feedback (RLHF) to align outputs with human preferences via algorithms such as Proximal Policy Optimization (PPO) (Schulman et al., 2017). Subsequent works proposed more advanced techniques such as Direct Preference Optimization (DPO) (Rafailov et al., 2023), SimPO (Meng et al., 2024), and SimPER (Xiao et al., 2025), which improve the stability and efficiency of RL training. More recently, researchers have explored both outcome-based rewards (Guo et al., 2025) and step-level rewards (Feng et al., 2025b) to improve problem-solving in domains such as mathematical reasoning (Guo et al., 2025), code generation (Chen et al., 2025b), and web retrieval (Wei et al., 2025; Lin et al., 2025b). In this work, we investigate RL for imperfect-information games, where sparse outcomes, hidden states, and adversarial dynamics make reward design particularly challenging. Our analysis shows that *both outcome-based and step-level RL signals are ineffective at improving LLMs' internal policies in poker*, motivating the use of solver-derived, regret-inspired signals as more reliable feedback.

## A.3  TOOL-INTEGRATED REASONING OF LLMs

Tool-integrated reasoning (TIR) has emerged as a promising approach to extend the capabilities of LLMs. Prior works demonstrate improvements in domains requiring precise computation or external knowledge, including mathematical calculation (Das et al., 2024), programming (Chen et al., 2022; Lin et al., 2026), and web search (Vu et al., 2023). Early studies such as PAL (Gao et al., 2023) prompt LLMs to generate code for execution, while ToRA (Gou et al., 2024) curate tool-use trajectories and apply imitation learning to train tool invocation. More recently, RL has been explored as an effective framework to improve TIR (Jin et al., 2025; Feng et al., 2025a; Zheng et al., 2025). For instance, Search-R1 (Jin et al., 2025) enables search-engine queries for QA, ReTool (Feng et al., 2025a) improves mathematical reasoning with a code sandbox, and DeepResearcher (Zheng et al., 2025) scales multi-hop retrieval and tool orchestration. Despite these advances, existing TIR re-

search largely targets fully observed, single-agent tasks. In contrast, poker involves stochasticity, hidden information, and adversarial dynamics, where tools must compute equilibrium-consistent strategies and counterfactual values rather than deterministic answers. To the best of our knowledge, ToolPoker is the first TIR framework for imperfect-information games. It integrates external poker solvers into LLMs, teaching them how to invoke solvers, and grounding their reasoning traces in solver outputs. This ensures rigorous, precise game-theoretic reasoning and GTO-consistent play, bridging prior works on strategic reasoning, RL, and TIR.

## B    BACKGROUND AND RULES OF POKER

In this section, we introduce the poker variants studied in our work. These games are widely used in the literature as benchmarks for imperfect-information reasoning because they balance tractability with the core challenges of hidden information, sequential decision-making, and stochasticity.

### B.1    KUHN POKER

Kuhn poker (Kuhn, 2016) is a minimalistic poker game designed to capture the essence of imperfect-information decision-making in a tractable form. The game is played with only three cards (e.g., Jack, Queen, King) and two players. Each player antes one chip, and a single betting round follows. Each player receives one private card, and the third card remains hidden.

Players can either check/bet (if no bet has been made) or call/fold (if a bet has been made). Because of its small size—only a handful of information sets—Kuhn poker admits closed-form solutions, including simple Nash equilibrium strategies that mix between bluffing with weak hands and value betting with strong hands. Despite its simplicity, it highlights the central strategic dilemma of poker: balancing deception and value extraction under hidden information.

### B.2    LEDUC HOLD'EM

Leduc Hold'em (Zaciragic et al., 2025) is a widely studied poker variant that extends Kuhn by introducing multiple betting rounds and public information. The game is played with a small deck of six cards consisting of two suits and three ranks. Each player antes one chip and receives a single private card. A first round of betting occurs, after which a single public card is revealed. A second round of betting then follows.

The addition of the public card dramatically increases strategic depth: players must update beliefs about opponents' ranges as new information is revealed, balance bluffing and value bets across streets, and plan actions that maximize long-term expected value. Although still small enough for exact or approximate equilibrium computation (e.g., via CFR (Zinkevich et al., 2007)), Leduc captures essential poker phenomena such as semi-bluffing, slow-playing, and range narrowing, making it a standard benchmark for algorithmic and LLM-based poker research.

### B.3    LIMIT TEXAS HOLD'EM

Limit Texas Hold'em (Bowling et al., 2015) is a more realistic and complex poker variant that is closely related to the full game of Texas Hold'em, which is the most popular poker format in practice. The deck consists of 52 standard playing cards. Each player is dealt two private hole cards, and up to five public community cards are revealed in stages: the flop (three cards), the turn (one card), and the river (one card). At each stage, players take turns acting in one of several betting rounds.

Unlike No-Limit Hold'em, bet sizes in Limit Hold'em are fixed and restricted to small or big bets depending on the round. Each hand therefore unfolds as a sequence of structured betting decisions, but the state space remains extremely large compared to Kuhn or Leduc. The presence of multiple streets, large range interactions, and complex pot-odds considerations make Limit Hold'em a significantly more challenging testbed for LLMs and reinforcement learning algorithms. Professional-level play in this environment demands mastery of equilibrium-based reasoning as well as opponent exploitation—skills that current LLMs struggle to replicate.

### B.4 Additional Details of Background and Preliminary

### B.5 Game-theoretic Reasoning

In poker, game-theoretic reasoning grounded in Nash Equilibrium is essential for professional-level play. A Nash Equilibrium represents a stable outcome in which each player's strategy is an optimal response to the others. Formally:

**Definition B.1** (Nash Equilibrium (Nash Jr, 1950)). *A Nash Equilibrium is a strategy profile in a game where no player can unilaterally improve their payoff by deviating from their current strategy, assuming the other players' strategies remain unchanged. Formally, a strategy profile $(a_1^*, a_2^*, \ldots, a_n^*)$ is a Nash Equilibrium if, for every player $i$:*

$$U_i(a_i^*, a_{-i}^*) \geq U_i(a_i, a_{-i}^*), \quad \forall a_i \in A_i \tag{5}$$

*where $A_i$ denotes the set of feasible actions for player $i$, $U_i$ is the utility function (expected payoff) of player $i$, and $a_{-i}^*$ represents the equilibrium strategies of all players other than $i$.*

Rather than relying solely on heuristics or pattern recognition, professional players systematically evaluate equity, ranges, and pot odds within a game-theoretic framework, thereby providing an optimal action. An illustrative example of such game-theoretic reasoning in practice is in Appendix B.6.

### B.6 Professional Players in Poker

To illustrate how professional poker players think, we provide a real example from the blog of a well-known Texas Hold'em professional player[1]. Unlike casual players who rely on intuition, professionals systematically evaluate a wide range of factors before acting, including:

- **Game context:** What are the stack sizes, pot size, and stack-to-pot ratio?
- **Ranges:** What range of hands should I continue with? What range does my opponent have? How does the board interact with these ranges, and which player benefits most?
- **Board texture and big hands:** Who holds the larger share of strong hands in this spot?
- **Mixed strategies:** What is my optimal mix between actions (e.g., 3-betting vs. calling, check-calling vs. check-raising)?
- **Bet sizing:** How many bet sizes do I need here (e.g., two sizes such as 30% pot and 90% pot)? Which size does my hand prefer relative to my overall range?
- **Randomization:** How do I randomize between actions to stay balanced (e.g., using a chip marker to decide frequencies)?
- **Opponent modeling:** What is my opponent's likely response to my bet? What physical tells, history, or reads do I have? At what strategic level are they operating, and what exploits should I consider?

This example shows that professional play is grounded in equilibrium-based reasoning, probabilistic mixing, and careful opponent modeling, far beyond heuristic or surface-level decision making.

Our behavior datasets are designed with these principles in mind, encouraging LLMs to reason through such questions. Details of the text-only BC dataset curation and TIR-enable BC dataset curation are provided in Appendix D.3 and Appendix G.2, respectively.

## C Additional Details of Preliminary Analysis in Sec. 3

### C.1 Comparison Methods

To comprehensively evaluate the performance of LLMs in playing poker, we consider both *traditional RL-based baselines* and *rule-based solver baselines*. RL methods serve as learning-based references that have been widely applied to imperfect-information games, while rule-based solvers provide near-equilibrium strategies that approximate ground truth. Specifically, we include the following methods:

---

[1]https://www.partypoker.com/blog/en/its-the-same-game-but-it-isnt.html

- NFSP (Heinrich & Silver, 2016): Neural Fictitious Self-Play is a pioneering framework for learning approximate Nash equilibria in imperfect-information games. It combines reinforcement learning to approximate best responses with supervised learning to approximate average strategies, enabling agents to learn directly from self-play experience.

- DQN Mnih et al. (2015): Deep Q-Network was one of the first breakthroughs in deep RL for sequential decision-making. Although originally designed for perfect-information environments such as Atari, subsequent works (Zha et al., 2021b; Guo et al., 2023) have adopted it as a baseline for imperfect-information games, including poker.

- DMC Zha et al. (2021b): The Deep Monte Carlo (DMC) algorithm is originally proposed for the Chinese card game DouDizhu. It leverages large-scale self-play with Monte Carlo policy optimization and demonstrates strong performance in complex imperfect-information card games. Following prior works (Zha et al., 2021b; Guo et al., 2023), we adapt DMC as a baseline for poker.

- CFR+ (Tammelin, 2014): Counterfactual Regret Minimization (CFR) (Zinkevich et al., 2007) is a foundational algorithm for solving imperfect-information games, converging to Nash equilibrium by iteratively minimizing counterfactual regret at each information set. CFR+ enhances CFR with linear regret updates and warm-start averaging, greatly accelerating convergence. It has become the de facto standard solver in large-scale poker domains and serves as a strong rule-based baseline in our evaluation.

- DeepCFR Brown et al. (2019): Building on CFR, DeepCFR employs neural function approximation to replace tabular regret tables, thereby generalizing across information sets. While CFR+ is provably effective, its computational cost grows prohibitively in large games such as Texas Hold'em. DeepCFR addresses this limitation by learning regret values via neural networks, making it applicable to larger domains and forming the basis of superhuman agents such as Libratus (Brown & Sandholm, 2019).

## C.2    CASE STUDIES OF LLMS' REASONING FLAWS

We provide the examples from Qwen2.5-3B and GPT-4o in Tab. 13 and 14 to illustrate why LLMs fail in playing poker. From these tables, we consistently observe three limitations of LLMs in playing poker: (1) Heuristic Reasoning; (ii) Factual Misunderstanding; and (iii) Knowing-Doing Gap. The detailed analysis of these case studies can be found in Sec. 3.3.

## C.3    EVALUATION METRICS OF THE LLM-AS-A-JUDGE FOR LLMS' REASONING

In the LLM-as-a-Judge approach used in quantitative analysis of LLMs' reasoning traces in Sec. 3.3, we use the following three metrics to validate the identified three reasoning flaws:

- **Heuristic Reasoning Score (HR)**: The judge prompt template is provided in Tab. 15.
- **Factual Alignment Score (FA)**: The judge prompt template is provided in Tab. 16.
- **Action-reasoning Consistency Score (AC)**: The judge prompt template is provided in Tab. 17.

## C.4    FULL DETAILS OF QUANTITATIVE ANALYSIS

To further validate the reasoning flaws observed in case studies, we adopt an LLM-as-a-Judge framework (Dubois et al., 2023). Specifically, we design three metrics: heuristic reasoning (HR), factual alignment (FA), and action–reasoning consistency (AC). Each generated reasoning trace is scored by three independent LLM judges on a 0–2 scale for each metric. GPT-4.1-mini (OpenAI, 2025) is used as the judge model. The metric definitions and judge prompts are in Appendix C.3.

From the table, we observe that (i) *Reasoning flaws persist across all models*. All evaluated LLMs demonstrate varying degrees of heuristic reasoning, factual misunderstanding, and knowing–doing gaps. For instance, Qwen2.5-3B obtains only 0.53 HR, 0.18 FA, and 1.53 AC, indicating weak factual grounding and limited strategic reasoning. Even the strongest model, o4-mini, while achieving the 1.80 HR, 1.56 FA, and 1.85 AC, still falls short of perfect action–reasoning consistency (1.85). This confirms that these flaws are systemic and persist across models. (ii) *Scaling improves but does not eliminate reasoning flaws*. Large and more powerful models, such as Qwen2.5-72B and o4-mini, generally achieve higher scores across all these metrics compared to their lightweight variants. This

suggests that increased scale and instruction tuning enhance the ability of LLMs to approximate game-theoretic reasoning and avoid factual mistakes. Nevertheless, the persistence of non-trivial gaps, particularly in FA and AC, indicates that scaling alone is insufficient to reach professional-level game-theoretic reasoning. (iii) *Action-reasoning consistency remains imperfect*. AC scores are stable across models (1.53–1.87) yet below the professional baseline of 2.0. Even the strongest model, o4-mini, reaches 1.85 but still shows knowing–doing gaps where reasoning diverges from action. To directly assess this, we compute mismatch proportions in Appendix C.5, which align with AC values and confirm it as both a valid proxy for and evidence of the *knowing–doing gap*.

## C.5 Human-in-the-Loop Evaluation for LLMs' Reasoning

To validate the reliability of LLM-based judging, we conduct a human-in-the-loop evaluation. Drawing on professional-style reasoning (Appendix B.6) and our behavior cloning prompt template (Appendix D.3), we use GPT-5 to curate 20 reasoning traces and have them scored by LLMs. These traces achieve perfect scores (all achieve maximum 2), showing strong alignment with human judgments, which we include as a reference for our analysis.

## C.6 Calibration and Validation of our LLM-as-a-Judge Score

In this subsection, we provide the details of how to calibrate and validate our LLM-as-a-Judge Score. **Judge calibration.** In Appendix C.3, we apply the LLM-as-a-Judge approach and use three metrics: *Heuristic Reasoning (HR)*, *Factual Alignment (FA)*, and *Action–reasoning Consistency (AC)* in the scale of 0-2. To calibrate this scale, we iteratively refined the HR/FA/AC rubrics and judge prompts using a small pilot set of representative hands.

- **General procedure.** We collect a small set of clearly *good*, *medium*, and *poor* reasoning traces for each dimension, manually assign target scores (0/1/2), and refine the textual criteria until the judge consistently reproduces the correct scores.

- **HR calibration.** We anchor the "0/1/2" rubric using examples that are (i) purely heuristic, (ii) partially grounded but inconsistent, and (iii) strongly aligned with game-theoretic principles (e.g., pot odds, range interactions).

- **FA calibration.** We provide objective poker quantities (equities, ranges, pot odds) from external solvers and instruct the judge to score *only factual correctness*.

- **AC calibration.** We explicitly instruct the judge to verify that the reasoning logically implies the same action as the final decision.

**Judge validation.** Following the protocol in Sec. 3.3, we manually curate 20 professional-style reasoning traces to use them and score them by LLMs. These traces achieve perfect scores (all achieve maximum 2), showing strong alignment with human judgments.

**Sensitivity and inter-rater LLM agreement.** Our LLM-as-a-Judge results in Tab. 2, Tab. 4, and Fig. 2 are consistent across two distinct poker environments (Leduc and Limit Hold'em), indicating that the judge is *not domain-sensitive*.

To further assess inter-rater agreement, we re-evaluate ToolPoker's Limit Hold'em reasoning traces using **GPT-5** as the judge (instead of the GPT-4.1-mini judge used in the main paper). All settings follow Section 5.3. The results are reported in Tab. 6. From the table, we observe close agreement between the two judge models, validating the robustness of our evaluation and reducing concerns about prompt sensitivity or model-specific bias.

Table 6: Inter-rater agreement: LLM-as-a-Judge scores (0–2) on ToolPoker's reasoning traces in Limit Texas Hold'em. We compare the original judge (GPT-4.1-mini) with another judge (GPT-5).

| Method | HR | FA | AC | Avg. |
|---|---|---|---|---|
| GPT-5 | 1.94 | 1.89 | 1.90 | 1.91 |
| GPT-4.1-mini | 1.93 | 1.92 | 1.94 | 1.94 |

# D FULL DETAILS OF BC-RIRL

## D.1 FULL DETAILS OF REGRET-INSPIRED RL FINE-TUNING

While BC helps LLMs imitate expert play, its limited dataset size and imitation-based nature make it insufficient for professional-level performance. As an initial attempt to refine policies beyond imitation, we explore a regret-inspired reinforcement learning (RIRL) framework. Prior approaches in both traditional RL (Heinrich & Silver, 2016; Zhao et al., 2022) and LLM-based RL (Guo et al., 2025) typically rely on outcome-based rewards (e.g., win/loss). However, in poker, especially in multi-round games such as Leduc Hold'em and Texas Hold'em—these sparse and noisy signals fail to capture the contribution of individual actions. To address this, we experiment with a step-level regret-guided reward that leverages signals from a pre-trained CFR solver, aligning fine-tuning with the principle that minimizing cumulative regret drives convergence to the Nash equilibrium.

**Regret-guided Reward Design**. Inspired by our analysis in Sec. 3.2, which highlights CFR as the state-of-the-art algorithm for approaching Nash equilibrium in imperfect-information games, we explore optimizing LLMs through regret minimization. Our key idea is to compute cumulative regrets with CFR and transform them into fine-grained reward signals that estimate each action's contribution. For a policy $\pi_\theta$ as player $i$, the cumulative regret of action $a_i^t$ at time $t$ is defined as:

$$r_t(a_i^t) = r_{t-1}(a_i^t) + I_t(a_i^t), \quad I_t(a_i^t) = u(\sigma_t^{a_i^t}, \sigma_t^{-a_i^t}) - u(\sigma_t), \tag{6}$$

where $\sigma_t$ denotes the strategy profile at time $t$, $\sigma_t^{-i}$ is the opponents' strategy, $u(\sigma_t)$ the expected utility under $\sigma_t$, and $u(\sigma_t^{a_i^t}, \sigma_t^{-i})$ is the utility when player $i$ deviates to action $a_i^t$. The instantaneous regret $I_t(a_i^t)$ measures how much better or worse $a_i^t$ performs relative to the current mixture strategy, while $R_t(a_i^t)$ aggregates this over time. To compare actions within the same decision point, we normalize regrets into a relative reward signal:

$$R(a_i^t) = \frac{R_t(a_i^t) - \text{mean}(\{r_t(a_j)\}_{j=1}^{|\mathcal{A}|})}{F_{\text{norm}}(\{r_t(a_j)\}_{j=1}^{|\mathcal{A}|})}, \tag{7}$$

where $F_{\text{norm}}$ denotes a normalization factor, chosen as the standard deviation in our implementation.

**Fine-tuning Objective**. Based on this signal, we fine-tune LLM policy via PPO (Schulman et al., 2017) with the following clipped RL objective:

$$\mathcal{L}_{\text{PPO}}(\theta) = -\mathbb{E}_{x \sim \mathcal{D}_s, y \sim \pi_{old}(\cdot|x)}$$
$$\left[ \min\left( \frac{\pi_\theta(y|x)}{\pi_{old}(y|x)} A, \text{clip}\left( \frac{\pi_\theta(y|x)}{\pi_{old}(y|x)}, 1-\epsilon, 1+\epsilon \right) \right) - \beta \mathbb{D}_{\text{KL}}(\pi_\theta(\cdot|c)||\pi_{ref}(y|x)) \right], \tag{8}$$

where $\pi_\theta$ and $\pi_{old}$ denote the current and previous policy models, respectively. $\epsilon$ is the clipping-related hyperparameter. $\pi_{ref}$ is the reference policy that regularizes $\pi_\theta$ update via a KL-divergence penalty, measured and weighted by $\mathbb{D}_{KL}$ and $\beta$, respectively. Generalized Advantage Estimation (GAE) (Schulman et al., 2015) is used as the advantage estimate $A$. $x$ denotes the input samples drawn from $\mathcal{D}$, which is composed of trajectories generated by the current policy $\pi_\theta$. $y$ is the generated outputs via policy LLMs $\pi_\theta(\cdot|x)$. The procedures of trajectory collection are detailed in Appendix D.4.

## D.2 FULL DETAILS OF COMPARISON RESULTS

We evaluate whether BC-RIRL improves LLMs' poker performance by fine-tuning Qwen2.5-7B and comparing against both traditional methods and vanilla LLMs. Results in Kuhn and Leduc Hold'em are reported in Tab. 3. We highlight three key findings: (i) *All RL-based fine-tuning variants improve performance in Kuhn Poker*. This suggests that both outcome-based and regret-guided feedback provide useful learning signals in simple environments with limited strategy space. (ii) *BC-RIRL surpasses direct prompting and BC-SPRL in Leduc Hold'em, though it still trails traditional algorithms such as CFR+*. For example, BC-RIRL gains 17.0 chips against GPT-4.1-mini, while still losing 34.0 chips against CFR+. This indicates that regret-guided dense feedback is more effective than sparse outcome-based rewards in complex tasks, but is sufficient to reach equilibrium-level play. (iii) *Pure RIRL without the BC stage does not yield improvements in Leduc Hold'em*. For instance, BC-RIRL and BC-SPRL gain $+17.0$ and $-64.5$ chips against GPT-4.1-mini, respectively.

This underscores the importance of BC in establishing a strong foundation of expert-like reasoning before RL fine-tuning.

To further assess whether BC-RIRL enhances reasoning quality, we adopt the LLM-as-a-Judge protocol from Sec. 3.3 and compute three reasoning-trace scores. Results in Leduc Hold'em and Limit Texas Hold'em are reported in Tab. 4, with additional case studies provided in Appendix F.1. Two findings are observed: (i) *RIRL consistently surpasses the baselines on HR and AC*. For example, BC-RIRL fine-tuned on Qwen2.5-7B reaches 1.93 HR and 1.90 AC in Leduc Hold'em, outperforming the strongest vanilla LLM, o4-mini, which achieves 1.80 HR and 1.85 AC. This shows that BC-RIRL effectively mitigates heuristic reasoning flaws and reduces the knowing–doing gap. (ii) *RIRL yields only marginal improvements in FA*. For instance, in Limit Texas Hold'em, BC-RIRL achieves 1.12 FA, only slightly higher than vanilla Qwen2.5-7B (0.87 FA) and still far behind o4-mini (1.65 FA). Together with the case studies, these results indicate that while BC-RIRL improves strategic reasoning and action–reasoning alignment, factual misunderstandings remain a notable challenge.

### D.3    ADDITIONAL DETAILS OF BEHAVIOR CLONING

We provide the BC data construct prompt template, which is shown in Tab. 18. GPT-5-mini is used as the target model for annotation. The detailed actions and other auxiliary quantities (e.g., winning probability and hand range) are obtained from a pre-trained CFR solver, equity calculator and other tools. These tools are implemented in Python.

### D.4    TRAJECTORY COLLECTION PROCEDURE

To collect trajectories for RL fine-tuning, we adopt an on-policy setting where the LLM policy competes against a random agent. At each iteration, the LLM plays a batch of $N$ games against the random agent ($N = 64$ in our setting). The LLM's actions from each round are stored as individual data samples. Formally, for an LLM policy $\pi_\theta$ with partial observation $o_i^t$ and action history $h_i^t$ at time step $t$, a sample is represented as $(o_i^t, h_i^t, a_i^t)$, where $a_i^t$ is the chosen action of player $i$. After each batch, the collected trajectories are used to fine-tune the LLM policy $\pi_\theta$, producing an updated policy $\pi_\theta'$ that is then used for subsequent data collection.

### D.5    IMPLEMENTATION DETAILS OF BC-RIRL

In the behavior cloning stage, we construct $5,000$ data samples with both reasoning traces and actions for behavior cloning. Specifically, to generate actions, we use CFR+ Tammelin (2014) to compete against a random player that randomly selects actions from the action space, and extract the actions from CFR+ as the ground-truth actions. The GPT-5-mini is then used to generate reasoning traces of these actions, where the prompt is provided in Appendix D.3. In the RL stage, we set Qwen2.5-7B-Instruct as the base model for fine-tuning.

## E    METHODOLOGY OF SPRL

Inspired by traditional RL in imperfect-information games (Heinrich & Silver, 2016; Zhang et al., 2024), we conduct $\delta$-uniform self-play by letting a single policy LLM $\pi_\theta$ play both sides. In each round, we (i) clone the current policy to obtain a fixed opponent $\pi_{\bar\theta}$; (ii) sample $N$ self-play games between $\pi_\theta(\cdot \mid f(o_1^t))$ and $\pi_{\bar\theta}(\cdot \mid f(o_2^t))$, alternating positions and random seeds, to collect trajectories $\mathcal{T}_\theta$; (iii) update $\pi_\theta$ with RL on $\mathcal{T}_\theta$ for $\delta$ steps while keeping $\pi_{\bar\theta}$ fixed; and (iv) refresh $\pi_{\bar\theta}$ with the latest $\pi_\theta$ to start the next cycle.

**Fine-tuning Objective**. To fine-tune LLMs via RL, we then formulate the RL objective function as follows:

$$\max_\theta \mathbb{E}_{x\sim\mathcal{D}_s, y\sim\pi_\theta(\cdot|x)}[r_\phi(x,y)] - \beta\mathbb{D}_{KL}[\pi_\theta(y|x)||\pi_{ref}(y|x)], \qquad (9)$$

where $\pi_\theta$ is the policy LLM being trained. $\pi_{ref}$ is the reference LLM (typically the initial pretrained LLM) that regularizes the policy update via a KL-divergence penalty, measured and weighted by $\mathbb{D}_{KL}$ and $\beta$, respectively. $x$ denotes the input samples drawn from $\mathcal{D}_s$, which is composed of trajectories generated by the current policy $\pi_\theta$ in a self-play setting. $y$ represents the generated

outputs via policy LLMs $\pi_\theta(\cdot|x)$. In this paper, we choose a commonly used Proximal Policy Optimization (PPO) (Schulman et al., 2017) as the backbone RL algorithm, which optimizes LLMs by maximizing the following objective:

$$
\mathcal{L}_{\text{PPO}}(\theta) = -\mathbb{E}_{x\sim\mathcal{D}_s, y\sim\pi_{old}(\cdot|x)}
$$
$$
\left[ \min\left( \frac{\pi_\theta(y|x)}{\pi_{old}(y|x)} A_{adv}, clip\left( \frac{\pi_\theta(y|x)}{\pi_{old}(y|x)}, 1-\epsilon, 1+\epsilon \right) A_{adv} \right) - \beta\mathbb{D}_{KL}(\pi_\theta(\cdot|c)||\pi_{ref}(y|x)) \right], \tag{10}
$$

where $\pi_\theta$ and $\pi_{old}$ denote the current and previous policy models, respectively. $\epsilon$ is the clipping-related hyperparameter. The advantage estimate $A_{adv}$ is computed using Generalized Advantage Estimation (GAE) (Schulman et al., 2015). We also investigate the performance of other commonly used RL algorithms, such as GRPO Shao et al. (2024).

**Reward Design**. Poker is a sequential decision-making task with multiple turns. The reward for player $i$ at time step $t$ is defined as the discounted cumulative return from $t$ until the end of the game:

$$
R_i^t = \sum_{k=t}^{T} \gamma^{k-t} r_i^k, \tag{11}
$$

where $\gamma \in (0, 1]$ is the discount factor balancing immediate and long-term outcomes. Because players only observe payoffs after a hand is completed, the task is characterized by sparse rewards: intermediate steps yield $r_i^k = 0$, while the terminal step provides $r_i^T$. We consider two types of terminal signals: (i) *binary outcome* reward, where $r_i^T = 1$ if the player wins the hand and $r_i^T = 0$ otherwise; and (ii) *normalized earnings* reward, where $r_i^T = c_{\text{earn}}/c_{\text{init}}$, with $c_{\text{earn}}$ the final net chip gain (or loss) and $c_{\text{init}}$ the initial chip count.

# F  Additional Details of Initial Attempt in Sec. 4

## F.1  Case Studies of BC-RIRL

We present case studies of Qwen2.5-7B fine-tuned with BC-RIRL in Leduc Hold'em (Tab. 19 and Tab. 20). The results show that after fine-tuning, the model can produce reasoning traces that resemble those of professional players. However, closer inspection reveals persistent factual misunderstandings. For example, the model claims that calling is the optimal CFR action, even though the prompt explicitly states that calling is not a legal move. This supports our conclusion in Sec. 4.3: while BC-RIRL improves action–reasoning consistency and professional-style imitation, factual inaccuracies remain a significant challenge, highlighting the limitations of BC-RIRL.

# G  Additional Detail of ToolPoker

## G.1  TIR Rollout Prompt Template

The TIR rollout prompt template for poker is provided in Tab. 21.

## G.2  TIR BC Reasoning Dataset Curation

To construct high-quality TIR data without incurring prohibitive annotation cost, instead of building a TIR reasoning-augmented dataset from scratch, we build an automated pipeline to programmatically augments the reasoning dataset from Sec. 4.1 with standardized tool invocation templates (e.g., `<tool></tool>`) and execution outputs (e.g., `<output></output>`). A detailed example of the appended tool invocation templates is provided in Tab. 22.

## G.3  Reward Design

Our hybrid reward function contain the following components:

- **Answer reward**: This reward enforces the alignment of LLMs' final action with the GTO-guarantee action from the CFR solver. Formally, given an LLM policy $\pi_\theta$ as the player $i$ with

---

**Algorithm 1** Fine-tuning Algorithm of ToolPoker for TIR.

---

**Input:** Policy model $\pi_\theta$, old policy $\pi_{\text{old}}$, task dataset $\mathcal{D}_t$, masking function $\mathcal{M}$
  **for** each training iteration **do**
    **for** each task $x$ in $\mathcal{D}_t$ **do**
      Sample ground-truth GTO action $\hat{a}$ of $x$
      Sample a rollout $y$ from $\pi_{\text{old}}$ for $x$:
      Initialize reasoning chain
      **while** not end of episode **do**
        Generate next segment: `<think>` or `<tool>`
        **if** tool is invoked **then**
          Interact with external poker solvers, obtain `<output>`
          Append output to reasoning chain
        **end if**
      **end while**
      Extract model-predicted action $a$ from final response $p$
      Compute the composite reward $R(a, \hat{a}, p)$
      Compute GAE advantages $\hat{A}$ for $y$
      Apply loss masking $\mathcal{M}$ to exclude tool output tokens
      Compute PPO loss $\mathcal{L}_{\text{PPO}}$ in Eq. 10 and update $\pi_\theta$
    **end for**
  **end for**

---

partial observation $o_i^t$ and action history $h_i^t$ at time step $t$, the answer reward is denoted as:

$$R_{\text{answer}}(a_i^t, \hat{a}_i^t) = \begin{cases} 1, & \text{if is\_equivalent}(a_i^t, \hat{a}_i^t), \\ -1, & \text{otherwise}, \end{cases} \tag{12}$$

where $a_i^t$ and $\hat{a}_i^t$ denote $\pi_\theta$'s predicted action and CFR solver's action at time step $t$. `is_equivalent`$(\cdot)$ checks whether the model's final action matches the CFR solver's action as the ground-truth action.

- **Format reward**: $R_{\text{format}}(\rho_i^t) \in \{0, 1\}$, which evaluates whether the reasoning trace follows the required structured schema with special tokens in the correct order: reasoning `<think></think>`, tool calling `<tool></tool>`, feedback output `<output></output>`, and final action `<answer></answer>`.

- **Tool execution reward**: $R_{\text{tool}}(\rho_i^t) = \text{Tool}_{\text{suc}}/\text{Tool}_{\text{tot}}$, which measures the fraction of successful tool calls in the reasoning trace, encouraging the model to invoke external tools effectively and integrate their outputs into subsequent reasoning.

### G.4 RL FINE-TUNING ALGORITHM FOR TIR

Alg. 1 summarizes the fine-tuning procedure of ToolPoker for enabling TIR in poker. Given a task dataset $\mathcal{D}_t$, where construction details are in Appendix G.5, the algorithm proceeds as follows.

For each task $x \in \mathcal{D}_t$ with a corresponding ground-truth action $\hat{a}$ from a CFR solver, we first obtain $G$ rollouts $y$ from the old policy $\pi_{\text{old}}$. Each rollout is generated step by step, where the model produces either a `<think>` segment (internal reasoning) or a `<tool>` call. If a tool is invoked, the model interacts with the external poker solver, retrieves the `<output>`, and appends it to the reasoning chain. This iterative process continues until the end of the episode. At the end of the rollout, we extract the model-predicted action $a$ from the final response $p$. A composite reward $R(a, \hat{a}, p)$ is then computed, combining answer accuracy, reasoning format, and tool-execution quality (see Appendix G.3). Using this reward, we estimate advantages $\hat{A}$ with Generalized Advantage Estimation (GAE) (Schulman et al., 2015). To ensure tool outputs do not dominate training, we apply a masking function $\mathcal{M}$ that excludes solver outputs from the loss. Finally, we compute the PPO loss $\mathcal{L}_{\text{PPO}}$ (Eq. 8) and update the policy $\pi_\theta$.

Through this iterative process, the model learns not only to query solvers for GTO-consistent actions and other auxiliary quantities but also to integrate solver outputs into coherent reasoning traces, thereby aligning action selection with rigorous game-theoretic principles.

## G.5 Implementation Details

We follow existing works (Feng et al., 2025a; Jin et al., 2025) to train ToolPoker with the VeRL (Sheng et al., 2024) framework. For RL fine-tuning, based on the existing work (Wang et al., 2025), we build an automated pipeline to curate an action-only dataset with $400,000$ samples for both Leduc Hold'em and Limit Texas Hold'em. Specifically, we use a pretrained CFR solver to compete against a random agent and collect the game states and actions of CFR to build such a dataset. Note that

Qwen2.5-7B-Instruct model is the base model. The max response length is set as $8,192$ tokens. The rollout model's temperature is $0.7$ and top-p is $0.6$. For behavior cloning, we curate a TIR dataset with $5,000$ samples with both actions and tool-integrated reasoning traces. During RL fine-tuning, the rollout batch size is set to $64$, and the mini update size is $16$. An AdamW optimizer is utilized with an initial learning rate $1e-6$.

## G.6 Additional Discussion

**Generalization without solvers**. In realistic settings, external tools may be unavailable or only intermittently accessible. To examine this, we ablate ToolPoker by removing RL fine-tuning and retaining only BC (Sec. 5.3). This variant shows weaker tool-use capability than full ToolPoker, and under intermittent tool access we find that HR and AC remain relatively high while FA degrades first. These results suggest that ToolPoker internalizes core strategic structures (e.g., range-based reasoning and mixed strategies), while solvers primarily supply precise numerical quantities—supporting our view that LLMs provide the reasoning framework whereas external tools ensure the accuracy of game-theoretic computations.

## G.7 Comparison with Existing Tool-use Framework

### G.7.1 Relation to Existing Tool-use Framework

While ToolPoker follows the general "LLM + tools" paradigm, it is designed specifically for imperfect-information poker games with game-theoretic principles, whereas prior frameworks focus on general tasks (e.g., math, QA, web search). This difference leads to several important challenges that make existing methods difficult to directly apply.

**Task difference: game-theoretic reasoning**. Prior TIR methods (Yao et al., 2022; Schick et al., 2023; Feng et al., 2025a) typically aim to obtain factual answers or execute deterministic API calls. In contrast, ToolPoker targets *strategic reasoning* in games where (i) the agent must reason under imperfect information, and optimal play requires *Nash-equilibrium (GTO) reasoning*, and (ii) explanations must reflect game-theoretic principles rather than surface-level logic. This setting requires multi-step strategic reasoning that goes substantially beyond previous tool-use scenarios.

**Existing frameworks cannot be directly adapted**. *Unstable interleaved reasoning and tool use.* Poker reasoning requires LLMs to generate game-theoretic explanations while coordinating multiple solver calls for diverse quantities (e.g., actions, equities, ranges). Directly applying a ReTool-style framework (Feng et al., 2025a) to teach LLMs to invoke multiple tools during reasoning would (i) force the model to call and integrate several specialized solvers for each hand, (ii) introduce error propagation from tool calls across multi-step game-theoretic reasoning trajectories, and (iii) lead to inaccurate explanations and degraded gameplay. *High data cost.* Toolformer-style approaches (Schick et al., 2023) usually require large-scale reasoning traces augmented with solver calls to fine-tune LLMs. For game-theoretic reasoning tasks, generating such traces demands expensive LLM annotation and careful domain-specific tool invocation, making it prohibitively costly to scale to expert-level poker play.

**ToolPoker: a design specifically addressing these challenges**. To overcome these issues, ToolPoker introduces two key design choices. *Equilibrium-oriented simplified interface.* Rather than asking the LLM to orchestrate multiple tools, ToolPoker consolidates all solver functionalities into a single API call that returns GTO actions as well as auxiliary quantities (e.g., equities, strategic ranges, hand distributions). This equilibrium-oriented interface stabilizes TIR–RL training and allows the LLM to focus on producing accurate, professional-level reasoning instead of managing complex tool orchestration. *Low-cost, expert-level TIR dataset.* Instead of relying on a large-scale

reasoning dataset, ToolPoker deliberately constructs a small curated expert reasoning dataset aligned with game-theoretic principles, and augments it with tool-calling templates and solver outputs. This provides a cost-efficient way to perform behavior cloning from expert-level play, followed by reinforcement learning fine-tuning.

### G.7.2 EMPIRICAL COMPARISON WITH RETOOL

We then empirically compare ToolPoker with ReTool Feng et al. (2025a) to validate the effectiveness of ToolPoker in our poker games with imperfect information. Specifically, we implement ReTool in Leduc Hold'em (same solver, same backbone LLM). We modify our BC dataset following the original ReTool protocol to teach the model to call multiple poker tools during reasoning, and keep the RL stage consistent with ReTool.

We compare both methods under the same settings as Section 5.3 using Qwen2.5-7B-Instruct. Gameplay and Reasoning results are shown in Tab. 7 and 8, respectively. These results show that while ReTool improves over prompting-only LLMs, ToolPoker achieves higher gameplay performance and expert-level reasoning quality, demonstrating the advantages of our simple but effective design in ToolPoker for game-theoretic reasoning tasks.

Table 7: Gameplay comparison results of ToolPoker and ReTool in Leduc Hold'em. Qwen2.5-7B-Instruct is the backbone model.

|  | NFSP | DQN | DMC | CFR+ | Avg. |
|---|---|---|---|---|---|
| Vanilla LLM | $-57.5$ | $-93.0$ | $-73.0$ | $-68.5$ | $-73.0$ |
| ReTool | $+5.5$ | $+8.5$ | $-4.0$ | $-8.0$ | $+0.5$ |
| ToolPoker | $\mathbf{+11.5}$ | $\mathbf{+18.0}$ | $\mathbf{+1.0}$ | $\mathbf{-3.0}$ | $\mathbf{+6.8}$ |

Table 8: Reasoning quality comparison results of ToolPoker and ReTool in Leduc Hold'em. Qwen2.5-7B-Instruct is the backbone model.

| Method | HR | FA | AC | Avg. |
|---|---|---|---|---|
| Vanilla LLM | 0.95 | 0.86 | 1.68 | 1.16 |
| ReTool | 1.84 | 1.65 | 1.88 | 1.79 |
| ToolPoker | $\mathbf{1.96}$ | $\mathbf{1.95}$ | $\mathbf{1.91}$ | $\mathbf{1.94}$ |

### G.8 IMPACT OF REWARD COMPONENT IN $R$

To study the contribution of each component in the composite reward, we implement three ablative variants of ToolPoker in Leduc Hold'em (Qwen2.5-7B-Instruct backbone), each removing one component from the composite reward in Eq. 4. All other settings follow Section 5.3. We report both gameplay performance and reasoning quality in Table 9 and 10. From these tables, we observe:

- $R_{\text{answer}}$ is the main driver of improvement. Removing it will make reasoning traces and final decisions less tightly aligned with solvers' outputs (e.g., GTO-consistent action), leading to worse gameplay performance and reasoning quality (e.g, AC).

- $R_{\text{format}}$ mainly stabilizes format and structure, with a smaller but positive effect on performance. Removing $R_{\text{format}}$ keeps gameplay competitive.

- $R_{\text{tool}}$ benefits reliable tool use. Removing it leads to a slight drop in gameplay performance and FA/AC scores.

### G.9 REWARD VISUALIZATION

We plot the per-component reward trajectories of ToolPoker in Leduc Hold'em in Fig. 3. Qwen2.5-7B-Instruct is the backbone model. From the figure, we can observe:

- $R_{format}$ and $R_{tool}$ rapidly approach near 1, indicating that the model can learn to produce correct formats and tool invocation quickly.

Table 9: Gameplay performance of ToolPoker and ablations in Leduc Hold'em. Qwen2.5-7B-Instruct is the backbone model.

| | NFSP | DMC | CFR+ |
|---|---|---|---|
| ToolPoker/$R_{\text{answer}}$ | $-58.5$ | $-72.0$ | $-54.5$ |
| ToolPoker/$R_{\text{format}}$ | $+11.5$ | $+1.0$ | $-4.0$ |
| ToolPoker/$R_{\text{tool}}$ | $+9.0$ | $+0.5$ | $-5.5$ |
| ToolPoker | $\mathbf{+11.5}$ | $\mathbf{+1.0}$ | $\mathbf{-3.0}$ |

Table 10: Reasoning quality metrics across ablations in Leduc Hold'em. Qwen2.5-7B-Instruct is the backbone model.

| Method | HR | FA | AC | Avg. |
|---|---|---|---|---|
| ToolPoker/$R_{\text{answer}}$ | 1.89 | 1.08 | 1.45 | 1.58 |
| ToolPoker/$R_{\text{format}}$ | 1.95 | **1.95** | **1.91** | 1.94 |
| ToolPoker/$R_{\text{tool}}$ | 1.95 | 1.89 | 1.87 | 1.90 |
| ToolPoker | **1.96** | **1.95** | **1.91** | 1.94 |

- $R_{answer}$ gradually increases over training with some variance but no signs of instability or collapse.

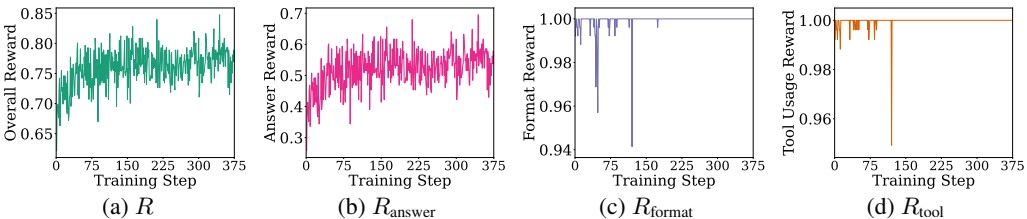

   (a) $R$       (b) $R_{\text{answer}}$      (c) $R_{\text{format}}$      (d) $R_{\text{tool}}$

Figure 3: Reward visualization of ToolPoker in Leduc Hold'em. Qwen2.5-7B-Instruct is the backbone model. (a) is the overall composite reward $R$, (b)-(d) shows the $R_{\text{answer}}$, $R_{\text{format}}$ and $R_{\text{tool}}$, respectively.

## H IN-DEPTH ANALYSIS OF TOOLPOKER

### H.1 TRANSFERABILITY & SCALABILITY

**Extending to Other Imperfect-information Games**. Although ToolPoker is empirically evaluated on poker in the main paper, the framework itself is not poker-specific. We choose poker as our primary testbed because it is a canonical benchmark for imperfect-information, game-theoretic reasoning: it has mature equilibrium solvers (e.g., CFR+), well-established evaluation protocols, and is widely used in prior works (Guo et al., 2023; Wang et al., 2025; Zhuang et al., 2025; Duan et al., 2024) to study strategic reasoning.

ToolPoker is architecturally game-agnostic and only requires access to a solver that, given a state description, returns equilibrium quantities (e.g., optimal actions, values, strategy distributions). To instantiate ToolPoker for another imperfect-information game, the required modifications are minimal:

- **Build solver API**. In a new game, collect required solvers for game-theoretic reasoning, and build a unified solver API that returns all supporting quantities from these solvers.
- **State encoding**. The game history, private information, and public observations of the new game must be encoded into text suitable for the LLM and for the unified solver API.
- **TIR reasoning dataset construction**. Similar to poker, we create a small-scale expert reasoning dataset containing high-quality reasoning traces augmented with solver outputs. This teaches the model how to read and interpret solver quantities and how to produce game-theoretic explanations.

- **Two-stage training pipeline**. We apply the same training procedure used in Section 5.2, which contains SFT on the solver-augmented reasoning dataset, followed by RL fine-tuning with our composite reward to refine tool-use behavior and action quality.

As an illustrative example, consider extending ToolPoker to an imperfect-information game, Mahjong. We would:

- encode each player's private hand, open melds, discards, and round context into text
- build an unified API such that the LLM can query this API to interface with external Mahjong solver to obtain actions (e.g., discard, call) and other supporting quantities (e.g., shanten count, tile-efficiency metrics, expected value, defensive risk), which are similar to equities and ranges in poker
- build a small solver-augmented reasoning set grounding explanations in strategic principles of Mahjong (e.g., tile efficiency, defense, hand value)
- apply the same two-stage training pipeline to finetune LLMs

**Empirical Results of ToolPoker in Extending to Three-player Leduc Hold'em**. To further demonstrate scalability, we adapted ToolPoker to a three-player Limit Texas Hold'em. We follow the steps above to fine-tune Qwen2.5-7B-Instruct using ToolPoker. We choose GPT-4.1-mini and vanilla Qwen2.5-7B-Instruct as the opponents, and compare the gameplay performance of the resulting model under the same settings in Section 5.3. The gameplay and reasoning quality results are reported in Tab. 11 and 12.

Table 11: Gameplay performance comparison across models in 3-player Leduc Hold'em.

| Qwen2.5-7B | GPT-4.1-mini | Qwen2.5-7B$_{\text{ToolPoker}}$ |
|---|---|---|
| $-36.7$ | $+5.9$ | $+\textbf{30.8}$ |

Table 12: LM-as-a-Judge score (0-2) evaluating reasoning traces of various LLMs in 3-player Leduc Hold'em.

| Method | HR | FA | AC | Avg. |
|---|---|---|---|---|
| Qwen2.5-7B | 0.93 | 0.88 | 1.60 | 1.14 |
| GPT-4.1-mini | 1.00 | 1.75 | 1.83 | 1.52 |
| Qwen2.5-7B$_{\text{ToolPoker}}$ | **1.93** | **1.90** | **1.88** | **1.90** |

From these tables, we observe that ToolPoker consistently outperforms vanilla LLM across both gameplay performance and expert-level reasoning scores in this new game, providing empirical evidence that ToolPoker generalizes beyond poker to other imperfect-information domains.

## H.2   ERROR ANALYSIS IN TOOLPOKER

In this subsection, we provide an in-depth error analysis of ToolPoker.

**Error patterns discussion.** As shown in Tab. 5, ToolPoker slightly underperforms CFR by 3 chips per 100 games, while still achieving comparable overall gameplay. To better understand this phenomenon, we conduct an error analysis and observe the following error patterns

- **State mis-specification**. The model may sometimes encode the game state (e.g., hand card, public card) imperfectly before querying the solver, which can lead to suboptimal actions and quantities from solvers.
- **Misalignment between solvers' outputs and final actions**. In some cases, the LLM may correctly receive solvers' outputs (e.g., action) but does not faithfully follow them in the final answer.

**Potential Mitigation**. To mitigate these errors, we consider several potential methods:

- **Additional faithfulness reward term**: Inspired by recent work on faithful agentic search (Xu et al., 2025), we can train a reward model to score how faithfully the reasoning aligns with solver outputs, and use this as an auxiliary reward during RL fine-tune.
- **Consistency-aware signal**: Similarly, we can add an auxiliary reward during RL fine-tuning to encourage correctly querying the external solvers with accurate states.

### H.3    ROBUSTNESS OF TOOLPOKER

**Robustness to noisy or human-style inputs.** A natural question is how ToolPoker handles inputs that deviate from clean CFR-style play, such as suboptimal, noisy, or human-generated trajectories. Although our main experiments rely on solver-labeled data, we emphasize that ToolPoker is already trained and evaluated in settings that include substantial off-equilibrium and non-expert behavior.

*(i) Training already includes noisy, off-equilibrium states.* As described in Appendix G.5, our RL dataset is constructed by letting a pretrained CFR agent play against a *random* opponent in both Leduc and Limit Texas Hold'em. We record all states but only use the CFR agent's actions as labels. Because the random agent frequently deviates from equilibrium play, the resulting trajectories contain diverse and imperfect state distributions far from idealized CFR self-play. Thus, ToolPoker is trained on a broad range of noisy, non-CFR game patterns rather than purely clean solver trajectories.

*(ii) Evaluation already involves diverse, non-expert opponents.* In online evaluation, ToolPoker plays against several traditional imperfect-information algorithms (NFSP, DQN, DMC) and LLM-based agents (e.g., prompting-only, BC+RIRL). These opponents generate highly variable and often non-equilibrium strategies. ToolPoker's consistent superiority across these settings demonstrates that it does not overfit to synthetic solver traces and can robustly respond to suboptimal or noisy play.

*(iii) Why ToolPoker is expected to generalize to human gameplay.* At inference time, ToolPoker does not rely on imitation of historical actions. Instead, it queries the unified solver API to retrieve equilibrium-oriented quantities (e.g., optimal action, equities, ranges) for the *current* state. Because solver outputs depend only on the observed state—regardless of whether the trajectory arose from CFR, heuristics, or human mistakes—ToolPoker can consistently anchor its reasoning to accurate game-theoretic guidance. This design inherently promotes robustness to out-of-distribution human-style inputs.

While we have not yet evaluated ToolPoker on real human gameplay, extending our assessment to human or crowd-sourced datasets is an exciting direction for future work.

## I    DISCUSSION OF FUTURE WORKS

Our research paves the way for further exploration of TIR in more complex strategic settings, shifting the focus beyond solely improving models' internal policies. Future work may explore richer tool ecosystems, multi-agent interactions, and principled frameworks for balancing internal reasoning with external computation, ultimately advancing the development of reliable AI systems for high-stakes decision making (Chen et al., 2024; Lin et al., 2025a; Chen et al., 2025a).

## J    LLM USAGE

We used an OpenAI LLM (GPT-5) as a writing and formatting assistant. In particular, it helped refine grammar and phrasing, improve clarity, and suggest edits to figure/table captions and layout (e.g., column alignment, caption length, placement). The LLM did not contribute to research ideation, experimental design, implementation, data analysis, or technical content beyond surface-level edits. All outputs were reviewed and edited by the authors, who take full responsibility for the final text and visuals.

Table 13: Realistic Examples of Qwen2.5-3B-Instruct in playing Limit Texas Hold'em. It demonstrates three limitations of LLMs in playing poker: (i) Heuristic reasoning; (ii) Factual Misunderstanding; (iii) Knowing-Doing Gap. Errors identified during reasoning are highlighted in red.

---

**Prompt**

You are a professional poker player playing 2-handed limit-holdem Poker. The following will be a game scenario and you need to make the optimal decision.

Here is a game summary:

In Limit Hold'em, each player receives exactly one private card, Small Blind and Big Blind ante 1 and 2 chips, respectively. Everyone started with 100 chips.
The player positions involved in this game are Small Blind, Big Blind.

In this hand:

Your position is Big Blind, and your holding is Your card: ['Two of Clubs', 'Ten of Hearts'].

Community card: Not yet revealed
Current betting round: pre-flop
Current pot: 6 chips

Your admissible actions:

fold
call
raise
Now it is your turn to make a move.

To remind you, the current pot size is 6 chips, and you are in position Big Blind, and your holding is Your card: ['Two of Clubs', 'Ten of Hearts'].

Decide on an action from the admissible actions based on the strength of your hand on this board, your position, and actions before you.

Your optimal action is:

---

**Model Output by Qwen2.5-3B-Instruct**

`<think>` Let's analyze the situation:
1. Position and Pot Size: You are in the Big Blind position, which is a disadvantageous position as you are forced to bet first. The pot size is 6 chips, and you have 50 chips in total, so you have a 90% stack to the pot.
2. Community Cards: The community cards are not yet revealed, so there is no information about the board yet.
3. Your Hand: You have the 'Two of Clubs' and 'Ten of Hearts'. This is a suited connector, which is a decent hand. However, it is not a strong hand by itself, and it is vulnerable to a flush draw or a straight draw.
4. Opponent's Hand: The opponent is in the Small Blind position, which is a more advantageous position as they can see your bet and respond accordingly. They have 50 chips as well, so they have a 90% stack to the pot.
5. Betting Rounds: This is the pre-flop round, and the only bet is the small blind. The big blind (you) is the first to act.
6. Admissible Actions: You can fold, call, or raise. Given the above considerations, let's evaluate the options:
- Fold: Folding would be a safe play if you believe your hand is not strong enough to compete with the opponent's potential hand. However, you have a decent hand, and the pot odds are favorable if you can get a raise.
- Call: Calling would mean you are matching the small blind's bet. This is a neutral play, but it doesn't give you any advantage or disadvantage.
- Raise: Raising would give you more information about your opponent's hand. If they call, you can see their hand and decide whether to continue or fold. If they fold, you win the pot with your hand. Considering the strength of your hand and the pot odds, a raise is the optimal action. You can raise to 4 chips, which is a reasonable amount to get your opponent to consider folding or calling. Therefore, the optimal action is to raise. `</think>`
`<answer>`raise`</answer>`

Table 14: Realistic Examples of GPT-4o in playing Limit Texas Hold'em. It demonstrates three limitations of LLMs in playing poker: (i) Heuristic reasoning; (ii) Factual Misunderstanding. Errors identified during reasoning are highlighted in red.

---

**Prompt**

You are a professional poker player playing 2-handed limit-holdem Poker. The following will be a game scenario and you need to make the optimal decision.

Here is a game summary:

In Limit Hold'em, each player receives exactly one private card, Small Blind and Big Blind ante 1 and 2 chips, respectively. Everyone started with 100 chips.
The player positions involved in this game are Small Blind, Big Blind.

In this hand:

Your position is Small Blind, and your holding is Your card: ['King of Spades', 'Ten of Clubs'].

Community card: Not yet revealed
Current betting round: pre-flop
Current pot: 3 chips

Your admissible actions:

fold
call
raise
Now it is your turn to make a move.

To remind you, the current pot size is 3 chips, and you are in position Small Blind, and your holding is Your card: ['King of Spades', 'Ten of Clubs'].

Decide on an action from the admissible actions based on the strength of your hand on this board, your position, and actions before you.

Your optimal action is:

---

**Model Output by GPT-4o**

`<think>` In heads-up Limit Hold'em pre-flop, hand values are generally stronger than in full-ring games because the range of hands played is wider and aggression is more frequent. The hand King-Ten offsuit is a marginal hand. It is not a premium hand, but it is playable in heads-up situations, especially from the Small Blind position where you have the advantage of acting first post-flop.

Given that the pot is 3 chips (small blind 1 chip + big blind 2 chips), and it is pre-flop with no bets yet, one might consider calling to see a flop. However, KT offsuit is not strong enough to profitably continue, and defending too wide risks being exploited.

Raising here is not justified, as the hand lacks sufficient value and fold equity. Folding avoids marginal spots and preserves chips for stronger hands.

Therefore, the optimal GTO decision is to call and see the flop.`</think>` `<anwer>`fold`</answewr>`

---

Table 15: Heuristic Reasoning (HR) Judge Prompt.

**HR Judge Prompt Template**

You are a judge evaluating the factual correctness of an LLM's reasoning in a poker decision.

You will be given:

(1) The game context (hand, position, pot size, betting round, etc.)

(2) The LLM's reasoning text

(3) The LLM's final action

(4) Objective equity analysis from an external tool (hand strength, win probability, Monte Carlo equity, etc.)

Your task:

- Identify if the reasoning shows reliance on *heuristics* (e.g., "I feel strong," "I have a face card," "opponent probably bluffing") rather than *strategic/game-theoretic principles* (e.g., pot odds, equity calculation, Nash equilibrium considerations).

- Score the reasoning on a 0–2 scale:

- 0 = Purely heuristic-driven (surface-level or intuitive analogies, no rigorous reasoning)

- 1 = Mixed (some heuristic reasoning, some strategic/game-theoretic reasoning)

- 2 = Principled (reasoning grounded mainly in sound game-theoretic or probabilistic principles)

Output format (JSON only):

{

"heuristic_reasoning_score": 0 | 1 | 2,

"explanation": "Brief explanation citing specific parts of the reasoning."

}

Table 16: Factual Alignment (FA) Judge Prompt Template.

**FA Judge Prompt**

You are a judge evaluating the factual correctness of an LLM's reasoning in a poker decision.

You will be given:

(1) The game context (hand, position, pot size, betting round, etc.)

(2) The LLM's reasoning text

(3) The LLM's final action

(4) Objective equity analysis from an external tool (win probability, hand range of both you and the opponent, etc.)

Your task:

- Compare the LLM's reasoning with the objective ground truth.

- Identify whether the reasoning contains factual misunderstandings, such as:

    • Incorrect classification of hand strength (e.g., calling AA "a weak hand")

    • Misstating probabilities or equity

    • Incorrect statements about positions, betting order, pot size, or available actions

    • Misinterpreting community cards, hole cards, or ranges

- Do not penalize the LLM for strategic differences (e.g., preferring raise vs. call), only for factual inaccuracies.

Scoring (0–2 scale):

- 0 = Major factual errors (core aspects wrong, e.g., misclassifying AA as weak, misstating betting order)

- 1 = Minor factual errors (some inaccuracies but overall interpretation mostly correct)

- 2 = Factually correct (no significant inaccuracies; reasoning aligns with objective equity and context)

Output format (JSON only):

{

"factual_correctness_score": 0 | 1 | 2,

"explanation": "Brief explanation citing specific parts of the reasoning."

}

Table 17: Action-reasoning Consistency Judge Prompt Template.

**AC Judge Prompt**

You are a judge evaluating the consistency between an LLM's articulated reasoning and its final decision in a poker hand.
You will be given:
(1) The poker state description (public cards, private cards, pot size, etc.)
(2) The LLM's step-by-step reasoning text
(3) The LLM's final action (fold, call, raise, etc.)
Your task:
- Determine whether the LLM's reasoning logically implies the same action as its final decision.
- If the reasoning suggests one action (e.g., fold) but the final action differs (e.g., call or raise), this is a "knowing–doing inconsistency."
- If the reasoning and action align, mark it as "consistent."
Scoring (0–2 scale):
- 0 = Inconsistent — The reasoning clearly points to one action, but the final decision is different.
- 1 = Partially consistent — The reasoning is mixed, ambiguous, or suggests multiple options, with the final action aligning with only part of the reasoning.
- 2 = Fully consistent — The reasoning unambiguously supports the final action and no contradictions are present.
Output format (JSON only):
{
"factual_correctness_score": 0 | 1 | 2,
"explanation": "Brief explanation of why reasoning matches, partially matches, or mismatches the final action."
}

Table 18: Behavior Cloning Dataset Construction Template.

| AC Judge Prompt |
| --- |

You are an poker expert working on {game_name} explaining optimal CFR-based play using rigorous game theory. Your task is to generate reasoning traces to explain the action from CFR from the game-theoretic perspective like professional poker players.

You will be given the following information:

Game: {game_name}
Street: {state_name};
Position: {position_name};
Stack (remaining chips):
- my stack: {my_stack},
- opponent stack: {opponent_stack};
Pot (betted chips): {total_pot}, my pot: {my_pot}, opponent pot: {opponent_pot};
Blinds/Antes: {antes};
Board/Public Card: {public card};
Action History: {action history};

My Winning Probability: {my_equity}
Opponent's Winning Probability: {opponent_equity}

My Hand Histogram: {my_hand_his}
Opponent's Hand Histogram: {opponent_hand_his}

Mix Action Strategy from CFR: {action_dist}

<think>
1) Situation — Summarize current situation from my perspective, e.g., position, stack,
2) Range Estimatation - Explain the range estimation of my hand and opponent's hand based on the hand histogram, respectively. And compare them to see what range of hand do I continue with?
3) Board Fit — Explain how the board texture interacts with both ranges, and whether future cards are more favorable for me or the opponent.
4) Hand Quality — Judge my hand's strength given its equity compared to opponent's hand, and also utilize the hand histogram to estimate who holds more of the strongest possible combos.
5) Strategy Mix — We are given the action probabilities from CFR, explain the strategy that which action is more likely to be taken to achieve equilibrium.
6) Stack Sizes/ Pot Size - Analyze the chips remaining and the pot size to see how much room there is to maneuver.
7) Opponent strategy - Analyze the action history to predict what is my opponent's strategy facing my bet.
8) Final Action - Based on the above analysis, explain the final action that I should take.
</think>

Be concrete, mathematical, and educational. Reference specific probabilities and regret values. Write clear numbered steps (1–8)

Format your response as:
<think>
[Detailed step-by-step strategic analysis with specific numbers]
</think>
<answer> {expert action} </answer>

Table 19: Case Study of BC-RIRL fine-tuned Qwen2.5-7B in Leduc Hold'em (Part I). Errors identified during reasoning are highlighted in red.

---

**Prompt**

---

You are an poker expert working on Leduc Hold'em (6 cards: 2J, 2Q, 2K in two suits) explaining optimal CFR-based play using rigorous game theory. Your task is to generate reasoning traces to explain the action from CFR from the game-theoretic perspective like professional poker players.

Leduc Hold'em is a simple two-player zero-sum imperfect-information poker game. The deck contains six cards: two Jacks (J), two Queens (Q), and two Kings (K). One private card is dealt to each player, and one card is placed aside unseen. There are two betting rounds: Pre-flop and Flop. In the Pre-flop stage, the community card is unseen. And in the Flop stage, a single community card is then revealed. In each round, the first player must bet or check. If a bet is made, the other player must call or fold or raise (up to one raise per round). If one player raises, the other player can only call the raise or fold. All bets are of fixed size. At showdown, a pair beats any high card; between high cards, K > Q > J.

Here is a game summary:

In unknown, each player receives exactly one private card, and small blind antes 1 chips and big blind antes 2 chips. Everyone started with 100 chips. The player positions involved in this game are Small Blind, Big Blind.

In this hand:

Your position is Small Blind, and your holding is Your card: **Queen of Heart**.

Community card: **King of Spade**
Current betting round: flop
Current pot: 4 chips

This is the historical action of the game:
Small Blind call; Big Blind check;

Your admissible actions:

fold
check
raise

Now it is your turn to make a move.

To remind you, the current pot size is 4 chips, and you are in position Small Blind, and your holding is Your card: Queen of Heart.

Decide on an action from the admissible actions based on the strength of your hand on this board, your position, and actions before you.

Your optimal action is:

---

**Model Output by** BC-RIRL **fine-tuned Qwen2.5-7B-Instruct (Part I)**

---

<think> 1) Situation & Game State:
  - Position: Small Blind (SB)
  - Current street: Flop
  - Pot size: 4 chips (SB 1 + BB 2 + previous actions)
  - Action history: SB called, BB checked
  - Effective stacks: 98 chips behind for both players
  - Your private card: Q$\heartsuit$
  - Board card: K$\spadesuit$

2) Hand Quality and Board Interaction:
  - You hold Q$\heartsuit$ while the board shows K$\spadesuit$.
  - Your hand strength is currently just high card (Queen), with potential to improve to QQ.
  - Because one King is already on the board, the chance opponent holds KK is reduced.
  - Relative to the board, your equity is moderate but not strong.

3) Strategy Mixing & CFR Output:
  - CFR equilibrium prescribes calling here with probability 1.0.
  - Calling is the unique optimal action.
  - Folding or raising has negative expected value.

---

Table 20: Case study of BC-RIRL fine-tuned Qwen2.5-7B in Leduc Hold'em (Part II), which is continued from Tab. 19. Errors identified during reasoning are highlighted in red.

---

**Model Output by** BC-RIRL **fine-tuned Qwen2.5-7B-Instruct (Part II)**

4) Stack Sizes & Future Commitment:
   - Pot-to-stack ratio is small (4 vs 98).
   - A call costs very little and preserves flexibility.
   - Raising would overcommit without justification.

5) Opponent Range Estimation:
   - BB checked, capping their range to many weak hands and some traps.
   - Against this range, equilibrium prefers a passive call.

6) Information Set & Regret Minimization:
   - CFR regret tables show positive cumulative regret only for calling.
   - Folding/raising have negative regret, thus dominated.

7) Conclusion:
   - The optimal action is to call.
   - This preserves equity and follows CFR equilibrium.
`</think>`
`<answer>`call`</answer>`

---

Table 21: TIR rollout structured prompt template.

---

**TIR Rollout Prompt**

You are a helpful AI Assistant that provides expert poker analysis. You are given a partial observation from a player and admissible actions, you will need to provide the GTO action to win the game. You now have the ability to invoke external poker solvers to obtain the GTO action and other auxiliary quantities (e.g., equity, all players' possible hand ranges, and the action distribution from CFR) during the reasoning. The tool calling will be wrapped in `<tool></tool>`. The returned results will be wrapped in `<output> </output>` to help provide precise game-theoretic reasoning with GTO final actions.

The reasoning process is wrapped in `<think> </think>` tags. The final action is wrapped in `<answer> </answer>`.

Note that:
- To invoke poker solvers, please write one JSON command inside `<tool> </tool>` tags. Each command should have a name and args key, with args being a dictionary.
- Example: `<tool>solver(params_name1 = params_value1, ...)</tool>`
- Tools expect specific JSON input formats . Do not make up tools or arguments that are not listed.
- After you have used the tools , you will see the tool outputs inside `<output> </output>` tags in the same order from the system.

For example, uou can respond in the following format:
`<think>`[Reasoning process]`</think>` `<tool>`[Poker solver execution arguments]`</tool>` `<output>`[Output of poker solver]`</output>` `<think>`[Reasoning process.]`</think>` `<answer>`[Final action]`</answer>`.

---

Table 22: Example of tool-integrated behavior cloning data

| Structured Prompt |
| --- |
| (omit the prompt description here) |

| Structured Output |
| --- |

```
<think>
```
1) Situation
- You are on the turn in Limit Hold'em acting from the Small Blind with stacks remaining 90 each and the pot currently 20 (you and opponent have each contributed 10). The board is S3, SJ, HQ, SK (three spades: 3♠, J♠, K♠; also J–Q–K high cards).
- The Big Blind checked to you on the turn.

Let's call poker solver to get GTO actions, equities and hand ranges. `<\think>`
```
<tool>solver(player_card=['SQ','C7'], public_card=['S3','SJ','HQ','SK'],
my_pot=6, opponent_pot = 6, my_raise_num=1, opponent_raise_num=1
legal_actions=['raise', 'fold', 'check'])
```

```
<think>
```
(omit reasoning process here) `</think>`
```
<answer>
```
call`</answer>`

