# OpenReview forum: "How Far Are LLMs from Professional Poker Players? Revisiting Game-Theoretic Reasoning with Agentic Tool Use"
_ICLR.cc/2026/Conference — ICLR 2026 Poster_

### Official Review · Reviewer_4eQy · 2025-10-24

**Soundness:** 3
**Presentation:** 3
**Contribution:** 2
**Rating:** 6
**Confidence:** 3

**Summary:**

This paper tackles poker as a domain for strategic reasoning and analyzes the shortcomings of LLMs on poker.
The authors examine what features of reasoning correlate with poor performance, finding gaps between action and knowledge and flawed or heuristic reasoning.
Models are compared to a range of baseline solvers.
To improve models, they propose a training approach consisting of behavior cloning on curated data followed by RL using PPO. They find that this improves performance over the untrained model, but that it still lags behind solvers like CFT.
To further improve the model, they propose ToolPoker, a method which enables LLMs to recruit external solvers and tools to improve.
They train ToolPoker also via a combination of BC and PPO and find that it further improves and closes much of the gap between models and CFP.

**Strengths:**

- The paper compares to standard baselines
- games are run across multiple runs
- both open and closed-source models are considered
- qualitative analysis of reasoning is backed by quantitative results
- The LLM judge is externally validated
- BC+RL results show improvements, with ToolPoker showing further improvements

**Weaknesses:**

- ToolPoker improvements are not that surprising to me. Basically the gains here seems to boil down to externalizing the parts of the task that are more difficult for the model to external solvers. Implementationally, the training method might be useful, but I don't see what research question it is addressing that hasn't already been addressed by the BC+RL experiments. It seems like in the end, ToolPoker is largely introducing an engineered system that would be of limited use, since it offloads most of the strategic reasoning the model would do to other tools (so no longer getting at the motivating point of the paper on strategic reasoning) while also being costlier and less effective than baseline poker-playing solutions.
- L083 the claim is made that ToolPoker is for imperfect-information games, but it is only evaluated on poker, and seems very explicitly engineered for poker with limited transfer to other games.
- The reasoning analysis does not take into account that reasoning might not be faithful. Plenty of prior work has called into question whether reasoning traces from LLMs are faithful explanations of their behavior (see https://aigi.ox.ac.uk/wp-content/uploads/2025/07/Cot_Is_Not_Explainability.pdf for references). The analysis seems to hinge on reasoning being a causal explanation of the model's actions -- if that's not the case, it would explain the knowledge-action gap.
- if ToolPoker uses CFR solver as a tool, why does it not outperform CFR in Table 5? L436 indicates it's a result of tool-calling errors, is there a way to reduce these?
- The comparison of tool calling reasoning feels spurious. In this case, the tools are providing a lot of information that was not available to the model without tools.

**Questions:**

Minor comments:
- Tables 1, 3, 4 are unreadably small

---

> ### Author Response · Authors · 2025-11-21
> **Response to Reviewer 4eQy (1)**
>
> We thank the reviewer for recognizing the systematic analysis, valid techniques, extensive experiments in our paper. Below we provide a point-by-point response to the reviewer's comments:
>
> > **W1: “ToolPoker gains are unsurprising; improvements come from outsourcing difficult reasoning to solvers. What research question does it address beyond BC+RL?”**
>
> **A**: **(i) “What research question does ToolPoker address beyond BC+RL?”**
>
> Thanks for this thoughtful question. We kindly clarify that the goal of ToolPoker is not merely to “offload” reasoning, but to investigate a distinct research question:
>
> Can LLMs, when equipped with external solvers, learn to produce **game-theoretic optimal (GTO)-consistent actions and precise game-theoretic reasoning in poker games with imperfect information?**
>
> This capability cannot be achieved by directly prompting vanilla LLMs or by BC+RL methods alone. Our **contributions and novelties** unfold in three steps:
>
> 1\. Systematic investigation of LLM limitations (Section 3).
> We conduct a systematical qualitative and quantitative study showing three persistent flaws in LLM poker reasoning, revealing three persistent flaws:
>
> * heuristic bias,
> * factual misunderstanding, and
> * knowing–doing gaps between reasoning and final actions.
>
> This analysis motivates the need for a framework that can let LLMs provide optimal actions and precise reasoning in poker.
>
> 2\. Initial attempt (BC+RIRL) shows internal fine-tuning is insufficient  (Section 4).
> To mitigate the aforementioned flaws, we first propose a method BC+RIRL to explore whether improving the LLMs’ internal policy alone is enough.
>
> * Gameplay: BC+RIRL improves over direct prompting but **still underperforms traditional algorithms like CFR+** (L316-321).
> * Reasoning quality: BC+RIRL helps LLMs imitate professional reasoning styles, they **continue to struggle with precise game-theoretic derivation(e.g., equities, range calibration)** (L316-321).
>
> These results reveal that **internal fine-tuning alone cannot yield both GTO-consistent actions and accurate expert-level reasoning**.
>
> 3\. ToolPoker fills this gap (Section 5).
> To bridge the gap of BC+RL, we then propose ToolPoker, which leverages LLM’s strength in tool use to empower LLMs to leverage external poker solvers to refine their actions and reasoning qualities.
>
> Our experimental results in Section 5.3 show that ToolPoker **substantially outperforms BC+RIRL, traditional algorithms, and prompting-based LLMs**, achieving both strong gameplay performance and professional-level reasoning.
>
>
> **(ii) ToolPoker is just an engineered system that outsources reasoning and is less effective than baseline poker-playing solutions.**
>
> We kindly clarify that ToolPoker is not simply outsourcing reasoning. Its contribution lies in enabling **LLM–solver coordination**, allowing the LLM to overcome its inherent weaknesses and produce high-quality, game-theoretic explanations and GTO-consistent decisions.
>
> Specifically, ToolPoker requires LLMs to learn
>
> * **when** a solver call is needed,
> * **how** to interpret structured solver outputs (equities, ranges, strategies), and
> * **how** to integrate these outputs into coherent multi-step reasoning and final decisions in poker.
>
> This is a nontrivial capability that neither traditional solvers nor existing LLM-based methods provide:
>
> * Traditional poker solvers such as CFR+ output GTO-consistent actions but provide **no human-readable reasoning**.
> * Prompted LLMs can generate explanations but produce **suboptimal decisions** and **exhibit all three flaws from Section 3.3** in their reasoning traces: (i) heuristic bias; (ii) factual misunderstanding and (iii) knowing-doing gaps.
> * **BC+RIRL** reduces heuristic bias and some knowing–doing gaps but still shows **severe factual misunderstanding** (L343–346).
>
> In contrast, our experiments in Section 5.3 show that ToolPoker:
>
> * achieves **state-of-the-art gameplay among LLM-based methods and traditional methods (e.g., NFSP and DQN)**, and **competitive performance with CFR**,
> * learns to call the solver strategically, and
> * generates **precise, logically consistent reasoning traces grounded in solver outputs**, addressing all three reasoning flaws (L437–446).
>
> The goal of ToolPoker is not to outperform CFR, but to leverage CFR’s GTO-consistent actions and other poker solvers’ output (e.g., hand equity, hand range) together for studying how LLMs can strategically integrate external tools to approach professional-level decision-making and reasoning closely aligned with game-theoretic principles.

---

> ### Author Response · Authors · 2025-11-21
> **Response to Reviewer 4eQy (2)**
>
> > **W2: L083 the claim is made that ToolPoker is for imperfect-information games, but it is only evaluated on poker.**
>
> **A**: Thanks for your insightful question. We kindly clarify that although ToolPoker is empirically evaluated on poker in the main paper, the framework itself is **not** poker-specific. We choose poker as our primary testbed because it is a **canonical benchmark for imperfect-information, game-theoretic reasoning**: it has mature equilibrium solvers (e.g., CFR+), well-established evaluation protocols, and is widely used in prior works \[3,4,5,6\] to study strategic reasoning.
>
> ToolPoker is architecturally game-agnostic and only requires access to a solver that, given a state description, returns equilibrium quantities (e.g., optimal actions, values, strategy distributions). To instantiate ToolPoker for another imperfect-information game, the required modifications are minimal:
>
> * **Build solver API**. In a new game, collect required solvers for game-theoretic reasoning, and build a unified solver API that returns all supporting quantities from these solvers.
> * **State encoding**. The game history, private information, and public observations of the new game must be encoded into text suitable for the LLM and for the unified solver API.
> * **TIR reasoning dataset construction**. Similar to poker, we create a **small-scale expert reasoning dataset** containing high-quality reasoning traces augmented with solver outputs. This teaches the model how to read and interpret solver quantities and how to produce game-theoretic explanations.
> * **Two-stage training pipeline**. We apply the same training procedure used in Section 5.2, which contains SFT on the solver-augmented reasoning dataset, followed by RL fine-tuning with our composite reward to refine tool-use behavior and action quality.
>
> As an illustrative example, consider extending ToolPoker to an imperfect-information game, **Mahjong**. We would
>
> * encode each player’s private hand, open melds, discards, and round context into text
> * build a unified API such that the LLM can query this API to interface with external Mahjong solvers to obtain actions (e.g., discard, call) and other supporting quantities (e.g., shanten count, tile-efficiency metrics, expected value, defensive risk), which are similar to equities and ranges in poker.
> * build a small solver-augmented reasoning set grounding explanations in strategic principles of Mahjong (e.g., tile efficiency, defense, hand value).
> * apply the same two-stage training pipeline to finetune LLMs
>
> To further demonstrate scalability, we adapted ToolPoker to a **three-player** Limit Texas Hold’em. We follow the steps above to fine-tune Qwen2.5-7B-Instruct using ToolPoker. We choose GPT-4.1-mini and vanilla Qwen2.5-7B-Instruct as the opponents, and compare the gameplay performance of the resulting model under the same settings in Section 5.3. The results are shown below:
>
> * Gameplay (**Table 10 in Appendix H.1**)
>
> | Qwen2.5-7B | GPT-4.1-mini | Qwen2.5-7B$\_{ToolPoker}$ |
> | :---- | :---- | :---- |
> | \-36.7 | \+5.9 | **\+30.8** |
>
> * Reasoning (**Table 11 in Appendix H.1**)
>
> | Method | HR | FA | AC | Avg. |
> | :---- | :---- | :---- | :---- | :---- |
> | Qwen2.5-7B | 0.93 | 0.88 | 1.60 | 1.14 |
> | GPT-4.1-mini | 1.00 | 1.75 | 1.83 | 1.52 |
> | Qwen2.5-7B$\_{ToolPoker}$ | **1.93** | **1.90** | **1.88** | **1.90** |
>
> ToolPoker consistently outperforms vanilla LLM across both gameplay performance and expert-level reasoning scores in this new game, providing empirical evidence that ToolPoker generalizes beyond poker to other imperfect-information domains.
>
> > **W3: The reasoning analysis does not take into account that reasoning might not be faithful. The analysis seems to hinge on reasoning being a causal explanation of the model's actions \-- if that's not the case, it would explain the knowledge-action gap.**
>
> **A**: Thanks for the thoughtful question. We acknowledge that LLMs’ chain-of-thought (CoT) reasoning traces are not guaranteed to be faithful.
>
> In our paper, we do **not** assume that the reasoning text is a causal explanation of the model’s internal decision process. In our analysis, reasoning traces and final actions are treated as **two observable output channels** of the same model, and our objective of the preliminary analysis in Section 3.3 is to evaluate the model’s *behavioral* capability in poker. Concretely, our “knowing–doing gap” is defined **operationally** as a mismatch between (i) the action advocated or implied in the reasoning text and (ii) the final action produced in \<answer\> for the same state. This definition depends solely on the observable divergence between these two outputs and does **not** require the reasoning to be faithful or causal.

---

> ### Author Response · Authors · 2025-11-21
> **Response to Reviewer 4eQy (3)**
>
> This discrepancy is a meaningful **behavioral inconsistency**, regardless of whether the CoT is fully faithful or partially post-hoc: it shows that the model fails to jointly produce actions and explanations that are self-consistent and aligned with game-theoretic principles. This is **one essential aspect of “thinking and playing” like professional players**.
>
> We will include this clarification and cite relevant literature \[2\] in our revised paper to make our analysis assumptions explicit.
>
> > **W4: If ToolPoker uses CFR solver as a tool, why does it not outperform CFR in Table 5? L436 indicates it's a result of tool-calling errors, is there a way to reduce these? It there a way to reduce these?**
>
> **A:** Thanks for the insightful question. We kindly clarify that, according to L436, the reason why ToolPoker slightly underperforms CFR is occasional errors in tool-use. To better understand this phenomenon, we conduct an error analysis and observe the following error patterns:
>
> * **State mis-specification**. The model sometimes may encodes the game state (e.g., hand card, public card) imperfectly before querying the solver, which can lead to suboptimal actions and quantities from solvers.
> * **Misalignment between solvers’ outputs and final actions**. In some cases, the LLM may correctly receive solvers’ outputs (e.g., action) but does not faithfully follow it in the final answer.
>
> Despite these errors, Table shows that ToolPoker still achieves **state-of-the-art gameplay performance among LLM-based methods and traditional algorithms (NFSP, DQN, DMC)**, and is **comparable with CFR.** This indicates ToolPoker has achieved large improvement over existing LLMs in producing high-quality actions and reasoning.
>
> To mitigate these errors, we consider several potential methods:
>
> * Additional faithfulness reward term: Inspired by recent work on faithful agentic search \[7\], we can train a reward model to score how faithfully the reasoning aligns with solver outputs, and use this as an auxiliary reward during RL fine-tune.
> * Consistency-aware signal: Similarly, we can add an auxiliary reward during RL fine-tuning to encourage correctly query the external solvers with accurate states.
>
> We will consider these as future directions of ToolPoker.
>
> > **W5: The comparison of tool calling reasoning feels spurious. In this case, the tools are providing a lot of information that was not available to the model without tools.**
>
> **A:** Thanks for your questions. We agree that the tool-augmented model does have strictly more information than the tool-free baseline. The difference is intentional: our preliminary analysis in Sections 3 reveal that LLMs suffer from three reasoning flaws: (i) heuristic bias, (ii) factual misunderstanding and (iii) knowing-doing gap. Our results in Section 4 further indicate that Improving models’ internal policies only slightly mitigates these issues, it
> remains insufficient for precise derivations or competitive gameplay, underscoring LLMs’ fundamental limitations in game-theoretic tasks.
>
> To address these gaps, Section 5 explores an alternative direction: leveraging LLMs’ strength in tool use so that they can incorporate external poker solvers to refine both actions and reasoning.
>
> To more fairly assess whether ToolPoker truly learns to *use* solver outputs (rather than merely having more information), we introduce a stronger baseline, **solver-based prompting**, which directly appends the solver’s outputs to the input prompt of a vanilla LLM and then asks it to generate reasoning traces and final actions. We compare ToolPoker with this baseline in Leduc Hold’em using Qwen2.5-7B-Instruct as the backbone, with other settings following Section 3.1. The LLM-as-a-Judge scores are:
>
> | Method | HR | FA | AC | Avg. |
> | :---- | :---- | :---- | :---- | :---- |
> | vanilla LLM | 0.95 | 0.86 | 1.68 | 1.16 |
> | solver-based prompting | 1.32 | 1.41 | 1.88 | 1.53 |
> | ToolPoker | **1.96** | **1.95** | **1.91** | **1.94** |
>
> From the table, we observe that solver-based prompting improves all three reasoning scores compared to the vanilla LLM without tool outputs, but still clearly underperforms ToolPoker. This shows that **simply providing solver information is not enough**: it only partially **alleviates the three reasoning flaws and remains far from producing high-quality, precise expert-level reasoning.** ToolPoker’s additional gains come from learning how to interpret and integrate solver outputs through our TIR \+ RL training, which validates ToolPoker’s effectiveness over just the availability of extra information.

---

> ### Author Response · Authors · 2025-11-21
> **Response to Reviewer 4eQy (4)**
>
> > **Q1: “Minor comments: Tables 1, 3, 4 are unreadably small”**
>
> **A:** Thanks for pointing that out. We have adjusted the table size in the revised version of our paper.
>
> We sincerely thank you again for your time and effort in improving our paper. We will include the above discussion in the revised version of our paper. If you have any further concerns or questions, please do not hesitate to let us know. We will respond to them timely.
>
> \[1\] Solving Large Imperfect Information Games Using CFR+
> \[2\] Chain-of-Thought Is Not Explainability
> \[3\] Suspicion-agent: Playing imperfect information games with theory of mind aware gpt-4.
> \[4\] Can Large Language Models Master Complex Card Games
> \[5\] PokerBench: Training Large Language Models to become Professional Poker Players
> \[6\] GTBench: Uncovering the Strategic Reasoning Limitations of LLMs via Game-Theoretic Evaluations
> \[7\] Beyond Correctness: Rewarding Faithful Reasoning in Retrieval-Augmented Generation.

---

> > ### Comment · Reviewer_4eQy · 2025-11-24
> > **Response to rebuttal**
> >
> > Thanks to the authors for the in-depth response and additional results on an additional poker variant. Given these responses and additional context for the positioning of the paper, I am updating my score to an 8.
> >
> > I understand that Mahjong would be hard to include in the time period of the rebuttal, but it would be great to see that extension for camera-ready.

---

> > > ### Author Response · Authors · 2025-11-24
> > >
> > > Thanks so much for your thoughtful follow-up and for updating your score. We greatly appreciate your positive assessment of our additional clarifications and your suggestion to extend ToolPoker to Mahjong in future work. We agree that this is an exciting direction, and we will explore this extension and include it in the camera-ready version if possible.
> > >
> > > If you have any further questions or suggestions, please do not hesitate to let us know. We will respond promptly.

---

### Official Review · Reviewer_4Bxf · 2025-11-02

**Soundness:** 3
**Presentation:** 3
**Contribution:** 2
**Rating:** 4
**Confidence:** 4

**Summary:**

This paper investigates the ability of LLMs to perform game-theoretic reasoning in the domain of two-player poker, a challenging incomplete-information game. The authors identify three fundamental reasoning flaws in current LLMs: reliance on shallow heuristics, factual misjudgments, and a marked "knowing-doing" gap between articulated reasoning and actions taken. Following attempts to mitigate these flaws using behavior cloning (BC) and regret-inspired reinforcement learning (RIRL), the authors propose ToolPoker in which LLMs interface with external solvers (e.g., CFR and equity calculators) via a unified tool API. ToolPoker combines imitation and RL to ensure both game-theoretic optimal (GTO) play and reasoning traces aligned with professional-level principles.

**Strengths:**

1. This paper is well-written and well-organized. The proposed method is simple and easy to follow.
2. This paper presents extensive experimental results with detailed analysis on the ablation study as well as limitations.

**Weaknesses:**

1. The novelty of the paper is limited. It mainly applies reinforcement learning to a large language model using a classic game-theoretic solver (e.g., CFR+) as the reward signal or direct PPO, without introducing any fundamentally new algorithmic contributions or insights. Essentially, the work repackages standard solver outputs within an RL fine-tuning framework, resulting in incremental rather than conceptual advancement. Also, while the related work section is decent, it omits several works that would be natural inclusions, such as [1]

2. While the core analysis and ToolPoker formulation are well-motivated by Leduc Hold'em and Limit Texas Hold'em, the current instantiation is restricted exclusively to these benchmarks. There is little empirical or conceptual discussion about scalability to larger, multi-player, or variable-rule settings.

3. The system occasionally produces factual misunderstandings when tool outputs are unavailable or misinterpreted, revealing a continuing challenge for robust factual alignment.

4. Most experiments are performed with synthetic datasets or CFR-solver-based action labels, raising questions about how well ToolPoker would generalize to noisy or inherently human gameplay traces.

Reference:

[1] Fine-Tuning Large Vision-Language Models as Decision-Making Agents via Reinforcement Learning

**Questions:**

1. Can the authors provide more details on how the LLM-as-a-Judge scores were calibrated and validated? Specifically, are there estimates of both inter-rater LLM agreement and alignment with human judges beyond the 20-trace reference set? How sensitive are reasoning scores in Tables 2 and 4 to the prompt, model, or domain?

2. How does ToolPoker handle non-standard, noisy, or human-style inputs that diverge from solver-generated play? Has the method been evaluated on datasets that originate from human-expert games or crowd-sourced play, and how robust are the results in such settings?

---

> ### Author Response · Authors · 2025-11-21
> **Response to Reviewer 4Bxf (1)**
>
> We thank the reviewer for recognizing the clear presentation, valid techniques, extensive experiments and detailed analysis in our paper. Below we provide a point-by-point response to the reviewer's comments:
>
> > **W1(i): The novelty of the paper is limited.**
>
> **A**: Thanks for your question. We kindly clarify that ToolPoker is only one part of our overall contribution. Beyond ToolPoker, our novelty and contributions also include (i) conducting a systematic analysis of LLMs’ limitations in poker and (ii) an initial attempt to improve LLMs via fine-tuning internal policies.
>
> **Contribution and novelty**
>
> Our paper makes three steps:
>
> 1\. Systematic investigation of LLM limitations (Section 3).
> We conduct a systematical qualitative and quantitative study showing three persistent flaws in LLM poker reasoning, revealing three persistent flaws:
>
> * heuristic bias,
> * factual misunderstanding, and
> * knowing–doing gaps between reasoning and final actions.
>
> This analysis motivates the need for a framework that can let LLMs provide optimal actions and precise reasoning in poker.
>
> 2\. Initial attempt (BC+RIRL) shows internal fine-tuning is insufficient  (Section 4).
> To mitigate the aforementioned flaws, we first propose a method BC+RIRL to explore whether improving the LLMs’ internal policy alone is enough.
>
> * Gameplay: BC+RIRL improves over direct prompting but **still underperforms traditional algorithms like CFR+** (L316-321).
> * Reasoning: BC+RIRL helps LLMs imitate professional reasoning styles, they **continue to struggle with precise game-theoretic derivation(e.g., equities, range calibration)** (L316-321).
>
> These results reveal that **internal fine-tuning alone cannot yield both GTO-consistent actions and accurate expert-level reasoning**.
>
> 3\. ToolPoker fills this gap (Section 5).
> To bridge the gap of BC+RL, we then propose ToolPoker, which leverages LLM’s strength in tool use to empower LLMs to leverage external poker solvers to refine their actions and reasoning qualities.
>
> Our experimental results in Section 5.3 show that ToolPoker **substantially outperforms BC+RIRL, traditional algorithms, and prompting-based LLMs**, achieving both strong gameplay performance and professional-level reasoning.
>
> **Relation to existing tool-use frameworks (e.g., ReAct \[1\], Toolformer \[2\] and ReTool \[3\])**
> While ToolPoker follows the general “LLM \+ tools” paradigm, **it is designed specifically for imperfect-information poker games with game-theoretic principles,** whereas prior frameworks focus on general tasks (e.g., math, QA, web search). This difference leads to several important challenges that make existing methods difficult to directly apply.
>
> 1\. Task difference: game-theoretic reasoning
> Prior TIR methods \[1,2,3\] usually aim to obtain factual answers or execute deterministic API calls. In contrast, ToolPoker targets **strategic reasoning** in games where:
>
> * The agent must reason under imperfect information and achieve optimal play requires **Nash-equilibrium (GTO) reasonin**g
> * Explanations must reflect game-theoretic principles not just surface-level logic.
>
> This requires **multi-step strategic reasoning** far beyond the scope of previous tool-use settings.
>
> 2\. Existing frameworks cannot be directly adapted
> (i) Unstable interleaved reasoning and tool use
> Poker reasoning requires LLMs to generate game-theoretic explanations while coordinating multiple solver calls for diverse quantities (e.g., actions, equities, ranges).
>
> Per L76-79, Directly apply **ReTool-style \[3\]** tool-use framework to teach LLMs to invoke multiple tools during reasoning would:
>
> * force LLMs to call and integrate several specialized solvers per hand,
> * causes tool-call error propagation across multi-step game-theoretic reasoning trajectories, and
> * leads to inaccurate explanations and degraded gameplay
>
> (ii) High data cost
> To make LLMs learn to think and play as professional players, Toolformer-style \[2\] approaches usually need to collect large-scale reasoning traces augmented with solver calls to fine-tune LLMs. In our game-theoretic reasoning tasks,  generating such data requires expensive LLM annotation and careful domain-specific tool invocation, making it prohibitively costly to scale.

---

> ### Author Response · Authors · 2025-11-21
> **Response to Reviewer 4Bxf (2)**
>
> **ToolPoker: A design specifically addressing these challenges**
> To overcome these issues, per L81-94, ToolPoker introduces:
>
> (i) Equilibrium-oriented simplified interface.
> Rather than asking the LLM to orchestrate multiple tools, ToolPoker consolidates all solver functionalities into a **single API call**, returning GTO actions and all auxiliary quantities (e.g., equities, strategic ranges, hand distributions).
>
> This interface **stabilizes TIR-RL training** and enables the LLM to focus on producing accurate, professional-level reasoning.
>
> (ii) Low-cost, expert-level TIR dataset.
> Instead of relying on a large-scale reasoning dataset, ToolPoker deliberately constructs a small curated expert reasoning dataset aligned with game-theoretic principles, and augments it with tool-calling templates and solver outputs.
>
> This provides a cost-efficient way for behavior cloning, followed by RL finetuning.
>
> **Empirical Comparison with ReTool (Appendix G.7)**
> Following your suggestion, we implement a **ReTool baseline** in Leduc Hold’em (same solver, same backbone LLM). We modify our BC dataset following the original ReTool protocol \[3\] to teach the model to call multiple poker tools during reasoning, and keep the RL stage consistent with ReTool.
>
> We compare both methods under the same settings as Section 5.3 using Qwen2.5-7B-Instruct. Results are shown below:
>
> * Gameplay (**Table 6 in Appendix G.7**)
>
> |  | NFSP | DQN | DMC | CFR+ | Avg.  |
> | :---- | :---- | :---- | :---- | :---- | :---- |
> | Vanilla LLM | \-57.5 | \-93.0 | \-73.0 | \-68.5 | \-73.0 |
> | ReTool | \+5.5 | \+8.5 | \-4.0 | \-8.0 | \+0.5 |
> | ToolPoker | **\+11.5** | **\+18.0** | **\+1.0** | **\-3.0** | **\+6.8** |
>
> * Reasoning (**Table 7 in Appendix G.7**)
>
> | Method | HR | FA | AC | Avg. |
> | :---- | :---- | :---- | :---- | :---- |
> | Vanilla LLM | 0.95 | 0.86 | 1.68 | 1.16 |
> | ReTool | 1.84 | 1.65 | 1.88 | 1.79 |
> | ToolPoker | **1.96** | **1.95** | **1.91** | **1.94** |
>
> These results show that while ReTool improves over prompting-only LLMs, **ToolPoker achieves higher gameplay performance and expert-level reasoning quality,** demonstrating the advantages of our simple but effective design in ToolPoker for game-theoretic reasoning tasks.
>
> > **W2: More empirical or conceptual discussion of ToolPoker’s scalability to large, multi-player, or variable-rule settings.**
>
> **A:** Thanks for your questions. We kindly clarify that our current experiments and analysis indeed mainly focus on three poker games: Kuhn, Leduc Hold’em and Limit Texas Hold’em, which are standard, non-trivial imperfect-information benchmarks.
>
> In addition, we would like to emphasize that **ToolPoker itself is architecturally game-agnostic** and can in principle be instantiated for more complex poker settings and other imperfect-information games. This only requires:
>
> * **Build solver API**. In a new game, collect required solvers for game-theoretic reasoning, and build a unified solver API that returns all supporting quantities from these solvers.
> * **State encoding**. The game history, private information, and public observations of the new game must be encoded into text suitable for the LLM and for the unified solver API.
> * **TIR reasoning dataset construction**. Similar to poker, we create a **small-scale expert reasoning dataset** containing high-quality reasoning traces augmented with solver outputs grounded in game-theoretic principles. This teaches the model how to read and interpret solver quantities and how to produce game-theoretic explanations.
> * **Two-stage training pipeline**. We apply the same training procedure used in Section 5.2, which contains SFT on the solver-augmented reasoning dataset, followed by RL fine-tuning with our composite reward to refine tool-use behavior and action quality.
>
> Under this abstraction, ToolPoker can, in principle, extend to **larger, multi-player, or variable-rule settings** with the help of specialized equilibrium solvers.
>
> To further demonstrate scalability, we adapted ToolPoker to a **three-player** Limit Texas Hold’em. We follow the steps above to fine-tune Qwen2.5-7B-Instruct using ToolPoker. We choose GPT-4.1-mini and vanilla Qwen2.5-7B-Instruct as the opponents, and compare the gameplay performance of the resulting model under the same settings in Section 5.3. The results are shown below:
>
> * Gameplay (**Table 10**)
>
> | Qwen2.5-7B | GPT-4.1-mini | Qwen2.5-7B$_{ToolPoker}$ |
> | :---- | :---- | :---- |
> | \-36.7| \+5.9 | **\+30.8** |
>
> * Reasoning (**Table 11**)
>
> | Method | HR | FA | AC | Avg. |
> | :---- | :---- | :---- | :---- | :---- |
> | Qwen2.5-7B| 0.93 | 0.88 | 1.60 | 1.14 |
> | GPT-4.1-mini| 1.00 | 1.75 | 1.83 | 1.52 |
> | Qwen2.5-7B$\_{ToolPoker}$ | **1.93**| **1.90** | **1.88** | **1.90** |
>
> ToolPoker consistently outperforms the baseline LLMs across both gameplay performance and expert-level reasoning scores in this three-player game, validating the scalability of ToolPoker to larger, multi-player, or variable-rule settings.

---

> ### Author Response · Authors · 2025-11-21
> **Response to Reviewer 4Bxf (3)**
>
> > **W3: The system occasionally produces factual misunderstandings when tool outputs are unavailable or misinterpreted, revealing a continuing challenge for robust factual alignment.**
>
> **A**: Thanks for the question. We kindly clarify that:
>
> **1\. While we still observe occasional factual misunderstandings in ToolPoker, they are much rare than in baselines. This is a substantial step toward more faithfully grounded game-theoretic reasoning.**
>
> As discussed in Section 3.3, vanilla LLMs frequently exhibit **factual misunderstanding** such as, mis-estimating hand strength or equity.
>
> ToolPoker is specially designed to address this issue by grounding game-theoretic reasoning in external poker solvers’ outputs (e.g., equity, GTO actions, hand range). While occasional factual mistakes still occur, our results in **Table 5** and **Figure 2** of our original paper show that:
>
> * ToolPoker achieves **state-of-the-art performance in producing high-quality actions and reasoning traces**
> * ToolPoker consistently obtains the **highest factual alignment (FA)** score among all LLM-based methods on both poker benchmarks.
>
> This indicates that ToolPoker is already a large step **toward more faithfully grounded game-theoretic reasoning**.
>
> **2\. Toward further improving factual alignment.**
>
> To future pushing robust factual alignment. One potential method is:
>
> * Additional faithfulness reward term (future work): Inspired by recent work on faithful agentic search \[4\], we can train a reward model to score how faithfully the reasoning aligns with solver outputs, and use this as an auxiliary reward during RL fine-tune.
>
> We will include the above discussion and the potential method in the revised version of our paper.
>
> > **W4 \&Q2: Most experiments use synthetic or CFR-labeled data. How would ToolPoker handle non-standard, noisy, or human-style inputs that diverge from solver-generated play? Has it been evaluated on human or crowd-sourced games?**
>
> **A**: Thanks for your question. We kindly clarify our current setting and why we expect ToolPoker to be reasonably robust.
>
> **1.Current training already includes noisy, off-equilibrium play.**
>
> As described in Appendix G.5, the RL dataset is built by letting a pretrained CFR solver **play against a random agent** in both Leduc and Limit Texas Hold’em. We record all game states but **only use the CFR actions** as labels.
>
> The random opponent induces many suboptimal, off-equilibrium trajectories, so the state distribution is already quite “noisy” and far from idealized CFR self-play. Thus, ToolPoker is trained on **a wide variety of non-CFR, imperfect play patterns** rather than purely clean solver trajectories.
>
> **2\. Evaluation already involves non-CFR, non-expert opponents.**
>
> In evaluation, ToolPoker is evaluated online against a diverse set of non-expert players, including traditional algorithms (e.g., NFSP, DQN, DMC) and LLM-based methods (e.g., direct prompting, BC+RIRL).
> These opponents generate **highly variable, non-equilibrium strategies.**
> ToolPoker’s consistent superiority across these settings shows that it does **not** **overfit to synthetic solver traces** and can handle noisy or suboptimal behaviors.
>
> **3\. Why we expect robustness to noisy or human traces.**
>
> At inference, ToolPoker does **not** imitate historical actions. Instead, it queries the unified solver API to obtain supporting quantities (e.g., actions, equities, and ranges) for *that* state.
>
> The solver’s **outputs depend only on the state**, **not on whether the trajectory from CFR or a human player.** Thus, ToolPoker can **always** produce high-quality actions and reasoning traces using external solvers, which makes it **naturally robust to noisy or human actions**.
>
> While we have not yet evaluated ToolPoker on real human gameplay, we consider this a direction for future work and include the above discussion.

---

> ### Author Response · Authors · 2025-11-21
> **Response to Reviewer 4Bxf (4)**
>
> > **Q1: Provide more details on how the LLM-as-a-Judge scores were calibrated and validated?**
>
> **A**: Thanks for your question. Below we clarify how our LLM-as-a-Judge scores are calibrated and validated.
>
> 1\. **Judge calibration.**
> As shown in Tables 8–10 (Appendix C), we use an LLM-as-a-Judge to assign 0–2 scores for **Heuristic Reasoning (HR)**, **Factual Alignment (FA)**, and **Action–reasoning Consistency (AC)**.
>
> To **calibrate** the 0–2 scale, we iteratively refined the HR/FA/AC rubrics and prompts on a small pilot set of hands:
>
> * **General procedure**:  we collect a small set of clearly *good*, *medium*, and *poor* reasoning traces for each dimension, hand-label them with target scores 01/2, and then adjust the textual criteria until the judge consistently assigned 0/1/2.
> * **HR calibration**: we anchored “0/1/2” using examples that are (i) purely heuristic, (ii) mixed, and (iii) clearly grounded in game-theoretic principles (e.g., pot odds, range reasoning).
> * **FA calibration**: we provide objective poker quantities (equity, hand ranges, pot odds, etc.) from external tools and instruct the judge to score **only factual correctness** of the reasoning
> * **AC calibration**: we explicitly instruct the judge to check whether the reasoning text logically implies the same action as the final decision
>
> **2\. Judge validation**
> To validate the judge, as in L216-218 in Section 3.3 of the original paper, we manually curate 20 professional-style reasoning traces to use them and score them by LLMs. These traces achieve perfect scores (all achieve maximum 2), showing strong alignment with human judgments.
>
> **3\. Sensitivity and “inter-rater” agreement**
> Our LLM-as-a-Judge results in Table 2, Table 4 and Figure 2 are across two poker environments (i.e., Leduc and Limit Hold’em), we observe similar average scores and relative rankings of all compared methods under HR/FA/AC. This indicates our reasoning scores are **not sensitive to the domain**.
>
> To better address your question on inter-rater LLM agreement, we re-evaluat ToolPoker’s reasoning traces on Limit Texas Hold’em using **GPT-5** as the judge model (instead of GPT-4.1-mini used in the paper). All other settings follow Section 5.3. The results are:
>
> | Method | HR | FA | AC | Avg. |
> | :---- | :---- | :---- | :---- | :---- |
> | GPT-5 | 1.94 | 1.89 | 1.90 | 1.91 |
> | GPT-4.1-mini | 1.93 | 1.92 | 1.94 | 1.94 |
>
> These show close agreement between the two judges, validating the inter-rater LLM agreement. The choose of strong judge models used in our paper also **reduce the risk of prompt sensitivity**.
>
> > **W1(ii): Miss related work [5].**
>
> **A:** Thanks for pointing that out. We have now added citation [5] and its discussion to the **Related Work** section (Section 6 & Appendix A.1) in the revised version of our paper.
>
> We sincerely thank you again for your time and effort in improving our paper. We will include the above discussion in the revised version of our paper. If you have any further concerns or questions, please do not hesitate to let us know. We will respond to them timely.
>
> \[1\] ReAct: Synergizing Reasoning and Acting in Language Models.
> \[2\] Toolformer: Language Models Can Teach Themselves to Use Tools.
> \[3\] ReTool: Reinforcement Learning for Strategic Tool Use in LLMs.
> \[4\] Beyond Correctness: Rewarding Faithful Reasoning in Retrieval-Augmented Generation.
> \[5\] Fine-Tuning Large Vision-Language Models as Decision-Making Agents via Reinforcement Learning.

---

### Official Review · Reviewer_whik · 2025-11-04

**Soundness:** 3
**Presentation:** 2
**Contribution:** 2
**Rating:** 4
**Confidence:** 3

**Summary:**

This paper investigates the strategic reasoning capabilities of LLMs in the domain of imperfect-information games, using poker as a rigorous and interpretable benchmark. The authors systematically evaluate several LLMs across realistic poker environments, including Leduc Hold’em and Limit Texas Hold’em, assessing both gameplay outcomes and the quality of reasoning traces. The results show that existing LLMs exhibit three recurring flaws. To address these deficiencies, the authors first test a two-stage internal improvement pipeline—behavior cloning followed by reinforcement learning with step-level rewards. Although this approach yields more coherent, human-like reasoning, it remains insufficient for accurate game-theoretic play. Motivated by these limits, the paper introduces ToolPoker, a TIR framework that enables LLMs to call external poker solvers for GTO actions and quantitative support such as equity and hand ranges. ToolPoker unifies solver interfaces through a single API, constructs an expert-level reasoning dataset augmented with solver outputs, and trains models using a combination of supervised fine-tuning and PPO-based reinforcement learning

**Strengths:**

- The first systematic study analyzing LLM reasoning and action alignment in poker, identifying fundamental weaknesses in heuristic dependence, factual errors, and knowing–doing gaps.

- A detailed investigation of whether behavior cloning and step-level RL can internally mitigate these flaws, revealing their limited capacity to achieve GTO-consistent reasoning.

- ToolPoker integrates external solvers into LLM reasoning for imperfect-information games, with a unified API and solver-augmented training corpus.

**Weaknesses:**

1. The composite reward (Eq. 4) combines R_answer, R_format, and R_tool with tunable weights.  How each component quantitatively contributes to tool-learning behavior. Providing ablation or sensitivity analyses—e.g., varying α_f and α_t, or visualizing reward trajectories—would improve transparency and reproducibility. Moreover, discussing how the model avoids reward hacking, e.g., overusing the solver or formatting cues without deeper reasoning, would strengthen the credibility of the RL setup.

2. ToolPoker’s underlying design—LLMs invoking structured external APIs and fine-tuned with PPO—is conceptually close to frameworks like ReAct (Yao et al., 2023), Toolformer (Schick et al., 2023), and ReTool (Feng et al., 2025). The paper would benefit from explicitly contrasting how ToolPoker extends these approaches to imperfect-information and equilibrium-seeking contexts, possibly through a comparative discussion table or controlled ablation with ReTool baselines. Without this, the contribution risks being perceived as an application-specific adaptation rather than a fundamentally new paradigm.

3. To reach the stated goal of establishing a “principled, general framework for tool-integrated strategic reasoning,” it would benefit from stronger theoretical justification. Why or when the hybrid architecture (LLM reasoning + solver calls) should converge toward a game-theoretic equilibrium, nor how tool-use uncertainty affects strategic optimality.  for instance, linking solver-invocation frequency to bounded rationality or expected regret

**Questions:**

1. Whether ToolPoker has any formal link to equilibrium convergence or regret minimization theory? Specifically, under what assumptions does the interaction between the LLM’s policy and the external solver guarantee or approximate GTO consistency?

2. How sensitive is the model’s performance to the weighting of the three reward components-answer, format, tool execution? Were any instabilities observed when tuning these hyperparameters?

3. Does ToolPoker include any mechanism to regulate when the solver should be called, or does it always invoke the solver at each decision step? If so, could you analyze cases where solver calls lead to redundant or conflicting information, and whether the model learns adaptive tool-use behavior over time?

4. The paper cites ReTool (Feng et al., 2025) but does not present a direct comparison. Given the conceptual similarity—both integrate external APIs into LLM reasoning—could the authors discuss key differences in design philosophy or experimental setup?

---

> ### Author Response · Authors · 2025-11-21
> **Response to Reviewer whik (1)**
>
> We thank the reviewer for recognizing our extensive analysis of LLMs in playing poker and valid technique details. Below, we provide a point-by-point response to the reviewer's comments:
>
> >**W1(i) & Q2: How each component of the composite reward quantitatively contributes to tool-learning behavior.**
>
> A: Thanks for your suggestion. In the revised version, we add (i) **ablation studies over reward components** and **(ii) visualizations of reward trajectories** in Appendix G.8 and G.9. Below we summarize the main findings.
>
> **1\. Ablation on reward components**
> We implement three ablative variants of ToolPoker in Leduc Hold’em (Qwen2.5-7B-Instruct backbone), each removing one component from the composite reward in Eq. (4):
>
> * ToolPoker/$R\_{answer}$: without answer reward
> * ToolPoker/$R\_{format}$: without format reward
> * ToolPoker/$R\_{tool}$: without tool execution reward
>
> All other settings follow Section 5.3. We report both gameplay performance and reasoning quality in the two tables below.
>
> * Gameplay (**Table 8 in Appendix G.8**)
>
> |  | NFSP | DMC | CFR+ |
> | :---- | :---- | :---- | :---- |
> | ToolPoker/$R\_{answer}$ | \-58.5 | \-72.0 | \-54.5 |
> | ToolPoker/$R\_{format}$ | \+11.5 | \+1.0 | \-4.0 |
> | ToolPoker/$R\_{tool}$ | \+9.0 | \+0.5 | \-5.5 |
> | ToolPoker | **\+11.5** | **\+1.0** | **\-3.0** |
>
> * Reasoning (**Table 9 in Appendix G.8**)
>
> | Method | HR | FA | AC | Avg. |
> | :---- | :---- | :---- | :---- | :---- |
> | ToolPoker/$R\_{answer}$ | 1.89 | 1.08 | 1.45 | 1.58 |
> | ToolPoker/$R\_{format}$ | 1.95 | **1.95** | **1.91** | **1.94** |
> | ToolPoker/$R\_{tool}$ | 1.95 | 1.89 | 1.87 | 1.90 |
> | ToolPoker | **1.96** | **1.95** | **1.91** | **1.94** |
>
> We observe:
>
> * $R\_{answer}$ is the main driver of improvement. Removing it will make reasoning traces and final decisions less tightly aligned with solvers’ outputs (e.g., GTO-consistent action), leading to worse gameplay performance and reasoning quality (e.g, AC).
> * $R\_{format}$  mainly stabilizes format and structure, with a smaller but positive effect on performance. Removing $R\_{format}$​ keeps gameplay competitive .
> * $R\_{tool}$ benefits reliable tool use. Removing it leads to a slight drop in gameplay performance and FA/AC scores.
>
> **2\. Reward trajectories and sensitivity**
> We also plot the **per-component reward trajectories** of ToolPoker during PPO training in **Figure 3 of** **Appendix G.9** of the revised paper. We can observe:
>
> * $R\_{format}$ and $R\_{tool}$​ **rapidly** approach near 1, indicating that the model can learn to produce correct formats and tool invocation quickly.
> * $R\_{answer}$​ gradually increases over training with some variance but **no signs of instability or collapse**.
>
> > **W1(ii): Discuss how the model avoids reward hacking, e.g., overusing the solver or formatting cues without deeper reasoning.**
>
> **A**: Thanks for the insightful question. We agree that this is an important concern and have designed ToolPoker to mitigate reward hacking:
>
> * **Low weights for format/tool rewards discourage shallow optimization**
>
> We deliberately assign *small* coefficients in the composite reward (Eq. 4): $\\alpha\_t$ \= $\\alpha\_f$ \= 0.5. Thus, it weakens the incentive to chase solver calls or formatting tricks.
>
> * **Unified solver API removes the incentive for repetitive solver calls**
>
> Rather than using a multi-tool calling pipeline, ToolPoker uses a single unified API call that returns all solver quantities. This simplifies the tool-use paradigm and ensures repeated calls do not produce additional information. Thus, LLMs have no incentive to “hack” the reward through excessive tool use.
>
> In addition, empirically, the reward trajectory visualizations in Appendix G.9 of our revised paper show that:
>
> * $R\_{format}$ and  $R\_{tool}$ quickly rise to near 1 then stay flat,
> * only $R\_{answer}$ continues to increase over time.
>
> This suggests that:
>
> * most of the learning signal over time comes from $R\_{answer}$;
> * the model is **not simply chasing the easiest sub-rewards but is improving decision quality**.
>
> We also include some **potential extensions** in our revised paper that could future reduce reward hacking in future work:
>
> * adding a **soft penalty** for repeated tool calls on the same state to **encourage high-quality tool calling rather than high-frequency** tool calling,
> * gradually decreasing $\\alpha\_t$​ and $\\alpha\_f$​ during RL finetuning so that later training becomes fully driven by $R\_{answer}$.

---

> ### Author Response · Authors · 2025-11-21
> **Response to Reviewer whik (2)**
>
> >**W2 & Q4: ToolPoker’s underlying design is conceptually close to frameworks like ReAct, Toolformer and ReTool. The paper would benefit from explicitly contrasting how ToolPoker extends these approaches to imperfect-information and equilibrium-seeking contexts, possibly through a comparative discussion table or controlled ablation with ReTool baselines.**
>
> **A:** Thanks for your insightful question.  We kindly clarify that ToolPoker is only one part of our overall contribution. Beyond ToolPoker, our novelty and contributions also include (i) conducting a systematic analysis of LLMs’ limitations in poker and (ii) an initial attempt to improve LLMs via fine-tuning internal policies.
>
> **Contribution & novelty of our paper**
>
> Our paper makes three steps:
>
> 1\. Systematic investigation of LLM limitations (Section 3).
> We conduct a systematical qualitative and quantitative study showing three persistent flaws in LLM poker reasoning, revealing three persistent flaws:
>
> * heuristic bias,
> * factual misunderstanding, and
> * knowing–doing gaps between reasoning and final actions.
>
> This analysis motivates the need for a framework that can let LLMs provide optimal actions and precise reasoning in poker.
>
> 2\. Initial attempt (BC+RIRL) shows internal fine-tuning is insufficient  (Section 4).
> To mitigate the aforementioned flaws, we first propose a method BC+RIRL to explore whether improving the LLMs’ internal policy alone is enough.
>
> * Gameplay: BC+RIRL improves over direct prompting but **still underperforms traditional algorithms like CFR+** (L316-321).
> * Reasoning: BC+RIRL helps LLMs imitate professional reasoning styles, they **continue to struggle with precise game-theoretic derivation(e.g., equities, range calibration)** (L316-321).
>
>
> These results reveal that **internal fine-tuning alone cannot yield both GTO-consistent actions and accurate expert-level reasoning**.
>
> 3\. ToolPoker fills this gap (Section 5).
> To bridge the gap of BC+RL, we then propose ToolPoker, which leverages LLM’s strength in tool use to empower LLMs to leverage external poker solvers to refine their actions and reasoning qualities.
>
> Our experimental results in Section 5.3 show that ToolPoker **substantially outperforms BC+RIRL, traditional algorithms, and prompting-based LLMs**, achieving both strong gameplay performance and professional-level reasoning.
>
> **Relation to existing tool-use frameworks (e.g., ReAct \[1\], Toolformer \[2\] and ReTool \[3\])**
> While ToolPoker follows the general “LLM \+ tools” paradigm, **it is designed specifically for imperfect-information poker games with game-theoretic principles,** whereas prior frameworks focus on general tasks (e.g., math, QA, web search). This difference leads to several important challenges that make existing methods difficult to directly apply.
>
> 1. Task difference: game-theoretic reasoning
>
> Prior TIR methods \[1,2,3\] usually aim to obtain factual answers or execute deterministic API calls. In contrast, ToolPoker targets **strategic reasoning** in games where:
>
> * The agent must reason under imperfect information and achieve optimal play requires **Nash-equilibrium (GTO) reasonin**g
> * Explanations must reflect game-theoretic principles not just surface-level logic.
>
> This requires **multi-step strategic reasoning** far beyond the scope of previous tool-use settings.
>
> 2. Existing frameworks cannot be directly adapted
>
> (i) Unstable interleaved reasoning and tool use
> Poker reasoning requires LLMs to generate game-theoretic explanations while coordinating multiple solver calls for diverse quantities (e.g., actions, equities, ranges).
>
> Per L76-79, Directly apply **ReTool-style \[3\]** tool-use framework to teach LLMs to invoke multiple tools during reasoning would:
>
> * force LLMs to call and integrate several specialized solvers per hand,
> * causes tool-call error propagation across multi-step game-theoretic reasoning trajectories, and
> * leads to inaccurate explanations and degraded gameplay
>
> (ii) High data cost
> To make LLMs learn to think and play as professional players, Toolformer-style \[2\] approaches usually need to collect large-scale reasoning traces augmented with solver calls to fine-tune LLMs. In our game-theoretic reasoning tasks,  generating such data requires expensive LLM annotation and careful domain-specific tool invocation, making it prohibitively costly to scale.

---

> ### Author Response · Authors · 2025-11-21
> **Response to Reviewer whik (3)**
>
> **Empirical Comparison with ReTool**
> Following your suggestion, we implement a **ReTool baseline** in Leduc Hold’em (same solver, same backbone LLM). We modify our BC dataset following the original ReTool protocol \[3\] to teach the model to call multiple poker tools during reasoning, and keep the RL stage consistent with ReTool.
>
> We compare both methods under the same settings as Section 5.3 using Qwen2.5-7B-Instruct. Results are shown below:
>
> * Gameplay (**Table 6 in Appendix G.7**)
>
> |  | NFSP | DQN | DMC | CFR+ | Avg.  |
> | :---- | :---- | :---- | :---- | :---- | :---- |
> | Vanilla LLM | \-57.5 | \-93.0 | \-73.0 | \-68.5 | \-73.0 |
> | ReTool | \+5.5 | \+8.5 | \-4.0 | \-8.0 | \+0.5 |
> | ToolPoker | **\+11.5** | **\+18.0** | **\+1.0** | **\-3.0** | **\+6.8** |
>
> * Reasoning (**Table 7 in Appendix G.7**)
>
> | Method | HR | FA | AC | Avg. |
> | :---- | :---- | :---- | :---- | :---- |
> | Vanilla LLM | 0.95 | 0.86 | 1.68 | 1.16 |
> | ReTool | 1.84 | 1.65 | 1.88 | 1.79 |
> | ToolPoker | **1.96** | **1.95** | **1.91** | **1.94** |
>
> These results show that while ReTool improves over prompting-only LLMs, **ToolPoker achieves higher gameplay performance and expert-level reasoning quality,** demonstrating the advantages of our simple but effective design in ToolPoker for game-theoretic reasoning tasks.
>
> > **W3\&Q1: To reach the stated goal of establishing a “principled, general framework for tool-integrated strategic reasoning,” it would benefit from stronger theoretical justification. Why or when the hybrid architecture (LLM reasoning \+ solver calls) should converge toward a game-theoretic equilibrium, nor how tool-use uncertainty affects strategic optimality. for instance, linking solver-invocation frequency to bounded rationality or expected regret**
>
> **A:**  Thank you for raising this important point. We kindly clarify that **ToolPoker does not aim to introduce a new equilibrium-convergent algorithm**, nor do we assume that the hybrid LLM–solver architecture will independently converge to a Nash equilibrium.
>
> Instead, in ToolPoker,  the **game-theoretic equilibrium comes from the external poker solvers (e.g., CFR solver),** which provides GTO-consistent actions, counterfactual values, range distributions, equities, etc. ToolPoker’s role is to train the LLM to correctly *interpret* and *integrate* these solver outputs within multi-step reasoning. Thus, we do not claim new theoretical guarantees beyond those already provided by the solver itself.
>
> A theoretical analysis of equilibrium convergence or regret minimization in the hybrid LLM-solver systems would require modeling how solver-call reliability and tool-use uncertainty propagate across multi-turn strategic reasoning, which is **beyond the scope of this paper**. We will regard this as an important avenue for future work.
>
> Although ToolPoker is not designed as a new equilibrium algorithm, it is *conditionally* equilibrium-consistent in a straightforward sense: **if** the solver returns equilibrium quantities and **if** the LLM reliably uses these outputs in its decisions, the induced policy inherits the solver’s equilibrium behavior. This then **provides a conceptual link to equilibrium reasoning**.
>
> > **Q3: Does ToolPoker include any mechanism to regulate when the solver should be called, or does it always invoke the solver at each decision step? If so, could you analyze cases where solver calls lead to redundant or conflicting information, and whether the model learns adaptive tool-use behavior over time?**
>
> **A:** We kindly clarify that in the current implementation of ToolPoker, we do not include a separate design to control when to call the solver. Instead, the solver is invoked at each decision step (each hand), and the unified API returns all relevant quantities (GTO action, equities, ranges, etc.) for that state.
>
> **1\. Conflict information**
> We do **not** observe conflicting information from solver calls. This is because all quantities (action, equity, ranges, values) are produced by a **single unified solver interface**, which ensures:
> * the outputs are internally consistent by construction, and
> * we do not observe cases where different solver outputs for the *same* state contradict each other in our experiments.
>
> This is also a key advantage of our unified interface compared with existing multi-tool use frameworks, where inconsistent responses can occur across heterogeneous tools.

---

> ### Author Response · Authors · 2025-11-21
> **Response to Reviewer whik (4)**
>
> **2\. Redundant information**
> We do observe some redundancy, which arises from repeated calling the unified API rather than inconsistent solver outputs:
>
> * In the multi-turn game-theoretic reasoning, the LLM may shift perspectives (e.g., analyzing pot odds, then opponent ranges).
> * When transitioning to a new reasoning subtask, the model sometimes **re-executes the same unified tool call**, even if parts of the required information were already returned earlier.
> * In these cases, the repeated solver call returns the same state-consistent quantities, resulting in redundancy, not inconsistency.
>
> **3\. Does the model earn adaptive tool-use behavior over time?**
> In the current version of ToolPoker, the unified API is available at every decision step (each hand), and the model is free to insert a \<tool\> call anywhere in its reasoning. During RL finetuning, the model learns that valid solver calls improve its overall reward, and in practice, it **typically calls the solver once per decision step** when it needs supporting quantities for game-theoretic reasoning.
>
> Thus, the RL stage mainly learns **how to interpret and use** the solver outputs, not **whether** to query the solver.
>
> We sincerely thank you again for your time and effort in improving our paper. We will include the above discussion in the revised version of our paper. If you have any further concerns or questions, please do not hesitate to let us know. We will respond to them timely.
>
> [1] ReAct: Synergizing Reasoning and Acting in Language Models.\
> [2] Toolformer: Language Models Can Teach Themselves to Use Tools.\
> [3] ReTool: Reinforcement Learning for Strategic Tool Use in LLMs.

---

> > ### Comment · Reviewer_whik · 2025-11-23
> >
> > Thanks for the thorough responses. I am not deeply familiar with this specific research area, and some of my earlier concerns arose from misconceptions on my side. After reading the clarifications and the newly added analyses, these have been resolved.
> >
> > I appreciate the effort the authors for addressing each point. I will update my score accordingly.

---

> > > ### Author Response · Authors · 2025-11-23
> > >
> > > Thanks so much for your thoughtful follow-up and for raising your score. We are glad that our clarifications helped address your questions. If you have any further questions, please do not hesitate to let us know. We will respond promptly.

---

### Official Review · Reviewer_3gDV · 2025-11-07

**Soundness:** 3
**Presentation:** 4
**Contribution:** 2
**Rating:** 8
**Confidence:** 3

**Summary:**

The paper uses poker to investigate game-theoretic reasoning in LLMs. It first compares vanilla LLM performance against several traditional baselines for game-theoretic reasoning, finding that LLMs noticeably underperform these baselines. An approach for improving LLM performance on poker is then introduced, with two main components: (1) behavior cloning, which consists of fine-tuning the LLM on expert-level trajectories augmented with reasoning traces and (2) RL fine tuning. This approach fails on its own, but these two high-level components are incorporated into the paper's last main contribution, ToolPoker, which allows the LLM to incorporate calls to external poker solvers into its reasoning trace (and is trained on a similar combination of behavior cloning and RL fine-tuning).

**Strengths:**

The paper is very clear in its presentation and generally high in quality: each newly introduced approach built on the last in a way that made the paper particularly easy to follow and made the motivation for the different components of the ToolPoker system apparent. The main points of significance and originality for the paper are that (1) it evaluates how LLMs reason about poker and presents qualitative and quantitative analyses of particular types of shortcomings in the reasoning process of vanilla LLMs and (2) applies a tool use framework to the poker setting.

**Weaknesses:**

The primary weakness of this paper lies in the novelty of the approach: while the paper does a very good job analyzing LLM performance on poker and explaining why the ToolPoker approach was developed, it is not clear if there is anything that sets ToolPoker apart from other tool-use frameworks, other than the task setting. In particular, explicitly comparing the strengths of ToolPoker with other approaches like ReTool (mentioned in the paper) would be helpful in evaluating the approach.

**Questions:**

1. What sets ToolPoker apart from other tool-use approaches?
2. Can ToolPoker be generalized to other imperfect-information games? What changes might have to be made (other than the tools called)?

---

> ### Author Response · Authors · 2025-11-21
> **Response to Reviewer 3gDV (1)**
>
> We sincerely thank the reviewer for the recognition of clear presentation, high quality, and the insight behind our analyses of LLM reasoning failures and the tool-use framework in poker. Below, we provide a point-by-point response to the reviewer's comments:
>
> > **W1\&Q1: What sets ToolPoker apart from other tool-use approaches (e.g., ReTool)?**
>
> **A**: Thanks for your question. We kindly clarify that ToolPoker is only one part of our overall contribution. Beyond ToolPoker, our novelty and contributions also include (i) conducting a systematic analysis of LLMs’ limitations in poker and (ii) an initial attempt to improve LLMs via fine-tuning internal policies.
>
> **Contribution & novelty of our paper**
>
> Our paper makes three steps:
>
> 1\. Systematic investigation of LLM limitations (Section 3).
> We conduct a systematical qualitative and quantitative study showing three persistent flaws in LLM poker reasoning, revealing three persistent flaws:
>
> * heuristic bias,
> * factual misunderstanding, and
> * knowing–doing gaps between reasoning and final actions.
>
> This analysis motivates the need for a framework that can let LLMs provide optimal actions and precise reasoning in poker.
>
> 2\. Initial attempt (BC+RIRL) shows internal fine-tuning is insufficient  (Section 4).
> To mitigate the aforementioned flaws, we first propose a method BC+RIRL to explore whether improving the LLMs’ internal policy alone is enough.
>
> * Gameplay: BC+RIRL improves over direct prompting but **still underperforms traditional algorithms like CFR+** (L316-321).
> * Reasoning: BC+RIRL helps LLMs imitate professional reasoning styles, they **continue to struggle with precise game-theoretic derivation(e.g., equities, range calibration, pot odds)** (L316-321).
>
> These results reveal that **internal fine-tuning alone cannot yield both GTO-consistent actions and accurate expert-level reasoning**.
>
> 3\. ToolPoker fills this gap (Section 5).
> To bridge the gap of BC+RL, we then propose ToolPoker, which leverages LLM’s strength in tool use to empower LLMs to leverage external poker solvers to refine their actions and reasoning qualities.
>
> Our experimental results in Section 5.3 show that ToolPoker **substantially outperforms BC+RIRL, traditional algorithms, and prompting-based LLMs**, achieving both strong gameplay performance and professional-level reasoning.
>
> **Relation to existing tool-use frameworks (e.g., ReAct \[1\], Toolformer \[2\] and ReTool \[3\])**
> While ToolPoker follows the general “LLM \+ tools” paradigm, **it is designed specifically for imperfect-information poker games with game-theoretic principles,** whereas prior frameworks focus on general tasks (e.g., math, QA, web search). This difference leads to several important challenges that make existing methods difficult to directly apply.
> 1\. Task difference: game-theoretic reasoning
> Prior TIR methods \[1,2,3\] usually aim to obtain factual answers or execute deterministic API calls. In contrast, ToolPoker targets **strategic reasoning** in games where:
>
> * The agent must reason under imperfect information and achieve optimal play requires **Nash-equilibrium (GTO) reasonin**g
> * Explanations must reflect game-theoretic principles not just surface-level logic.
>
> This requires **multi-step strategic reasoning** far beyond the scope of previous tool-use settings.
>
> 2\. Existing frameworks cannot be directly adapted
> (i) Unstable interleaved reasoning and tool use
> Poker reasoning requires LLMs to generate game-theoretic explanations while coordinating multiple solver calls for diverse quantities (e.g., actions, equities, ranges).
>
> Per L76-79, Directly apply **ReTool-style \[3\]** tool-use framework to teach LLMs to invoke multiple tools during reasoning would:
>
> * force LLMs to call and integrate several specialized solvers per hand,
> * causes tool-call error propagation across multi-step game-theoretic reasoning trajectories, and
> * leads to inaccurate explanations and degraded gameplay
>
> (ii) High data cost
> To make LLMs learn to think and play as professional players, Toolformer-style \[2\] approaches usually need to collect large-scale reasoning traces augmented with solver calls to fine-tune LLMs. In our game-theoretic reasoning tasks,  generating such data requires expensive LLM annotation and careful domain-specific tool invocation, making it prohibitively costly to scale.

---

> ### Author Response · Authors · 2025-11-21
> **Response to Reviewer 3gDV (2)**
>
> **ToolPoker: A design specifically addressing these challenges**
> To overcome these issues, per L81-94, ToolPoker introduces:
>
> (i) Equilibrium-oriented simplified interface.
> Rather than asking the LLM to orchestrate multiple tools, ToolPoker consolidates all solver functionalities into a **single API call**, returning GTO actions and all auxiliary quantities (e.g., equities, strategic ranges, hand distributions).
>
> This interface **stabilizes TIR-RL training** and enables the LLM to focus on producing accurate, professional-level reasoning.
>
> (ii) Low-cost, expert-level TIR dataset.
> Instead of relying on a large-scale reasoning dataset, ToolPoker delibrately construct a small curated expert reasoning dataset aligned with game-theoretic principles, and augments it with tool-calling templates and solver outputs.
>
> This provides a cost-efficient way for behavior cloning, followed by RL finetuning.
>
> **Empirical Comparison with ReTool (Appendix G.7)**
> Following your suggestion, we implement a **ReTool baseline** in Leduc Hold’em (same solver, same backbone LLM). We modify our BC dataset following the original ReTool protocol \[3\] to teach the model to call multiple poker tools during reasoning, and keep the RL stage consistent with ReTool.
>
> We compare both methods under the same settings as Section 5.3 using Qwen2.5-7B-Instruct. Results are shown below:
>
> * Gameplay (**Table 6 in Appendix G.7**)
>
> |  | NFSP | DQN | DMC | CFR+ | Avg.  |
> | :---- | :---- | :---- | :---- | :---- | :---- |
> | Vanilla LLM | \-57.5 | \-93.0 | \-73.0 | \-68.5 | \-73.0 |
> | ReTool | \+5.5 | \+8.5 | \-4.0 | \-8.0 | \+0.5 |
> | ToolPoker | **\+11.5** | **\+18.0** | **\+1.0** | **\-3.0** | **\+6.8** |
>
> * Reasoning (**Table 7 in Appendix G.7**)
>
> | Method | HR | FA | AC | Avg. |
> | :---- | :---- | :---- | :---- | :---- |
> | Vanilla LLM | 0.95 | 0.86 | 1.68 | 1.16 |
> | ReTool | 1.84 | 1.65 | 1.88 | 1.79 |
> | ToolPoker | **1.96** | **1.95** | **1.91** | **1.94** |
>
> These results show that while ReTool improves over prompting-only LLMs, **ToolPoker achieves higher gameplay performance and expert-level reasoning quality,** demonstrating the advantages of our simple but effective design in ToolPoker for game-theoretic reasoning tasks.
>
> > **Q2: Can ToolPoker be generalized to other imperfect-information games? What changes might have to be made (other than the tools called)?**
>
> **A**: Yes. ToolPoker is architecturally game-agnostic and only requires access to a solver that, given a state description, returns equilibrium quantities (e.g., optimal actions, values, strategy distributions). To instantiate ToolPoker for another imperfect-information game, the required modifications are minimal:
>
> * **Build  solver API**. In a new game, collect required solvers for game-theoretic reasoning, and build a unified solver API that returns all supporting quantities from these solvers.
> * **State encoding**. The game history, private information, and public observations of the new game must be encoded into text suitable for the LLM and for the unified solver API.
> * **TIR reasoning dataset construction**. Similar to poker, we create a **small-scale expert reasoning dataset** containing high-quality reasoning traces augmented with solver outputs. This teaches the model how to read and interpret solver quantities and how to produce game-theoretic explanations.
> * **Two-stage training pipeline**. We apply the same training procedure used in Section 5.2, which contains SFT on the solver-augmented reasoning dataset, followed by RL fine-tuning with our composite reward to refine tool-use behavior and action quality.
>
> As an illustrative example, consider extending ToolPoker to an imperfect-information game, **Mahjong**. We would
>
> * encode each player’s private hand, open melds, discards, and round context into text
> * build a unified API such that the LLM can query this API to interface with external Mahjong solvers to obtain actions (e.g., discard, call) and other supporting quantities (e.g., shanten count, tile-efficiency metrics, expected value, defensive risk), which are similar to equities and ranges in poker.
> * build a small solver-augmented reasoning set grounding explanations in strategic principles of Mahjong (e.g., tile efficiency, defense, hand value).
> * apply the same two-stage training pipeline to finetune LLMs.

---

> ### Author Response · Authors · 2025-11-21
> **Response to Reviewer 3gDV (3)**
>
> To further demonstrate scalability, we adapted ToolPoker to a **three-player** Limit Texas Hold’em. We follow the steps above to fine-tune Qwen2.5-7B-Instruct using ToolPoker. We choose GPT-4.1-mini and vanilla Qwen2.5-7B-Instruct as the opponents, and compare the gameplay performance of the resulting model under the same settings in Section 5.3. The results  are shown below:
>
> * Gameplay (**Table 10 in Appendix H.1**)
>
> | Qwen2.5-7B | GPT-4.1-mini | Qwen2.5-7B$\_{ToolPoker}$ |
> | :---- | :---- | :---- |
> | \-36.7 | \+5.9 | **\+30.8** |
>
> * Reasoning (**Table 11 in Appendix H.1**)
>
> | Method | HR | FA | AC | Avg. |
> | :---- | :---- | :---- | :---- | :---- |
> | Qwen2.5-7B | 0.93 | 0.88 | 1.60 | 1.14 |
> | GPT-4.1-mini | 1.00 | 1.75 | 1.83 | 1.52 |
> | Qwen2.5-7B$\_{ToolPoker}$ | **1.93** | **1.90** | **1.88** | **1.90** |
>
> ToolPoker consistently outperforms vanilla LLM across both gameplay performance and expert-level reasoning scores in this new game, providing empirical evidence that ToolPoker generalizes beyond poker to other imperfect-information domains.
>
> We sincerely thank you again for your time and effort in improving our paper. We will include the above discussion in the revised version of our paper. If you have any further concerns or questions, please do not hesitate to let us know. We will respond to them timely.
>
> [1] ReAct: Synergizing Reasoning and Acting in Language Models. \
> [2] Toolformer: Language Models Can Teach Themselves to Use Tools.\
> [3] ReTool: Reinforcement Learning for Strategic Tool Use in LLMs.

---

### Author Response · Authors · 2025-11-28
**General summary of rebuttal response**

Dear Reviewers and Area Chair,

We sincerely thank all reviewers for their valuable feedback and thoughtful comments on our submission. We are pleased to see recognition of the key strengths of our work, and **appreciate that the reviewers who responded during the discussion period confirmed that our rebuttal fully resolved their concerns**:

* **Reviewer 3gDV**: the clear presentation, high quality, and the insight behind our analyses of LLM reasoning failures and the proposed tool-use framework in poker.
* **Reviewer whik**: the extensive analysis of LLMs in playing poker and valid technique details.
* **Reviewer 4Bxf**: the clear presentation, valid techniques, extensive experiments and detailed analysis.
* **Reviewer 4eQy**: the systematic analysis, solid techniques, and comprehensive experimental evaluation.

> ## **Contribution & novelty of our paper**

We would like to summarize **the main contributions of our work, which are highlighted by the reviewers and further elaborated in our rebuttal**. Importantly, our novelty and contributions include:

* conducting a systematic analysis of LLMs’ limitations in poker;
* an initial attempt to improve LLMs via fine-tuning internal policies; and
* a tool-use framework that is specifically designed for imperfect-information games, grounded in game-theoretic principles.

**1\. Systematic investigation of LLM limitations (Section 3).**
We conduct a systematical qualitative and quantitative study showing three persistent flaws in LLM poker reasoning, revealing three persistent flaws:

* heuristic bias,
* factual misunderstanding, and
* knowing–doing gaps between reasoning and final actions.

This analysis motivates the need for a framework that can let LLMs provide optimal actions and precise reasoning in poker.

**2\. Initial attempt (BC+RIRL) shows internal fine-tuning is insufficient  (Section 4).**
To mitigate the aforementioned flaws, we first propose a method, BC+RIRL, to explore whether improving the LLMs’ internal policy alone is enough.

* Gameplay: BC+RIRL improves over direct prompting but **still underperforms traditional algorithms like CFR+** (L316-321).
* Reasoning: BC+RIRL helps LLMs imitate professional reasoning styles, they **continue to struggle with precise game-theoretic derivation(e.g., equities, range calibration)** (L316-321).

These results reveal that **internal fine-tuning alone cannot yield both GTO-consistent actions and accurate expert-level reasoning**.

**3\. ToolPoker fills this gap (Section 5).**
To bridge the gap of BC+RL, we then propose ToolPoker, which leverages LLMs’ strength in tool use to empower LLMs to leverage external poker solvers to refine their actions and reasoning qualities.

Our experimental results in Section 5.3 show that ToolPoker **substantially outperforms BC+RIRL, traditional algorithms, and prompting-based LLMs**, achieving both strong gameplay performance and professional-level reasoning.

> ## **Additional strengths recognized by reviewers and clarified in the rebuttal**

We also summarize several key questions that were initially raised by the reviewers and that, after our rebuttal and revisions, were acknowledged as clarified and viewed as strengths of the paper.

1. **ToolPoker novelty (Appendix G.7)**. We **clearly clarify the differences and novelty of ToolPoker relative to existing tool-use frameworks** and include an additional experiment directly comparing ToolPoker with ReTool to support our claims.
2. **Generalization to other imperfect-information games (Appendix H.1)**. We **clarify the generalization of ToolPoker to other imperfect-information games** by discussing how ToolPoker can be extended to another imperfect-information game, Mahjong, and by conducting an experiment on three-player Limit Texas Hold’em.

We have incorporated these discussions into the revised version of our paper.

> ## **Reviewers were satisfied with our rebuttal and increased their scores (average 6.5; November 23-24)**

During the discussion period, two reviewers, **whik** and **4eQy**, responded to our rebuttal and stated that our clarifications had fully addressed their concerns. They also indicated that they had raised their scores: **whik increased the score from 4 to 6 on November 23**, and **4eQy** **updated the score from 6 to 8 on November 24**. Our latest scores during the discussion period were therefore **8, 6, 4, and 8 (avg. 6.5)**. We greatly appreciate these positive assessments and are glad that the additional analyses and explanations in the rebuttal helped resolve the reviewers’ questions.

Overall, we are very grateful for the reviewers’ engagement and constructive feedback, which allowed us to further strengthen our work. We believe these strengths, combined with the revisions and additional experiments, demonstrate the significance and impact of our contributions.

Thanks for your time and consideration.

Best regards,

The Authors

---

> ### Author Response · Authors · 2025-11-29
> **Additional summary to assist the Area Chair’s evaluation**
>
> >## **Reviewers’ follow-ups acknowledged effective rebuttal and improved assessments**
>
> Before the discussion freeze (**Nov. 23-24**), two reviewers explictly confirmed that our clarification and new experiments fully addressed their earlier questions. At that time, the paper’s scores were **(8, 8, 6, 4)** with an **average score of 6.5**:
>
> * **Reviewer 3gDV** provided a **positive initial review (score 8)** emphasizing the clarity, high quality, and insightfulness of our analyses on LLM reasoning failures and the validity of ToolPoker.  We submitted additional clarifications and experiments to address their suggestions/questions, though the reviewer did not provide a follow-up comment before the discussion freeze.
>
> * **Reviewer whik** initially gave a score of 4, mainly raising questions regarding ToolPoker’s distinction from existing tool-use frameworks and its theoretical justification. In our rebuttal, we provided detailed clarifications and new experiments comparing with ReTool, along with an expanded discussion of the theoretical grounding. **On Nov. 23, the reviewer responded**, acknowledging that **“some of my earlier concerns arose from misconceptions on my side; after reading the clarifications and the newly added analyses, these have been resolved.”**. The reviewer then **raised the score from 4 $\rightarrow$ 6**.
>
> * **Reviewer 4Bxf** initially assigned a score of 4, focusing on novelty and scalability. In our rebuttal, we provided additional clarifications and experiments addressing novelty, and we also **extended ToolPoker to another imperfect-information game (Mahjong) and added experiments on a three-player Limit Texas Hold’em setting**. While this reviewer **did not provide a follow-up comment** before the discussion freeze, **Reviewers whik and 4eQy** explicitly acknowledged that our responses **sufficiently addressed these concerns**.
>
> * **Reviewer 4eQy gave a positive initial review (score 6)**, and confirmed that our rebuttal fully resolved their questions. They stated: **”Thanks to the authors for the in-depth response and additional results on an additional poker variant. Given these responses and additional context for the positioning of the paper, I am updating my score to an 8.”** The reviewer **raised the score from 6 $\rightarrow$ 8 on Nov. 24**, prior to discussion freeze.
>
> >## **Our review address reviewers' questions point-by-point**
>
> **Our rebuttal addressed the reviewers’ questions point by point.** Below, we summarize the key clarifications and revisions incorporated into the revised version of our paper during the discussion:
>
> * **Appendix G.7:  Relation to existing tool-use frameworks.**
>
> We expand the discussion comparing ToolPoker with existing tool-use frameworks [1,2,3], emphasizing the unique challenges of imperfect-information, game-theoretic settings and how ToolPoker addresses them. We also add a new **empirical comparison between ToolPoker and ReTool** in Leduc Hold’em, validating ToolPoker's effectiveness.
>
> * **Appendix G.8 & G.9: Composite Reward $R$**
>
> We add an ablation study on the composite reward $R$, and include per-component reward trajectory plots for ToolPoker.
>
> * **Appendix H.1: Transferability and scalability**
>
> We discuss how ToolPoker extends to other imperfect-information games, using **Mahjong** as an example. We also introduce a new experiment on **three-player Limit Texas Hold’em**, where ToolPoker consistently outperforms baseline models in both gameplay and reasoning metrics, demonstrating scalability.
>
> * **Appendix H.2: Analysis of ToolPoker vs. CFR and error analysis**
>
> We add further analysis comparing ToolPoker with CFR and include an error study in ToolPoker. We also discuss potential mitigations.
>
> * **Minor revisions**
>
> We adjust the sizes of Table 1, 3, 4.
>
> **These clarifications and revisions were acknowledged by the reviewers who responded to our rebuttal and subsequently increased their ratings**.
>
> [1] ReAct: Synergizing Reasoning and Acting in Language Models.\
> [2] Toolformer: Language Models Can Teach Themselves to Use Tools.\
> [3] ReTool: Reinforcement Learning for Strategic Tool Use in LLMs.

---

### Meta-Review · Area_Chair_rkgR · 2026-01-06

**Summary:**

This paper systematically studies LLMs’ game-theoretic reasoning via poker, identifying three core flaws (heuristic bias, factual misunderstandings, "knowing–doing" gaps) and proposing ToolPoker—a tool-integrated framework that leverages external solvers for GTO-consistent actions and professional reasoning. Experiments show ToolPoker achieves state-of-the-art gameplay and aligns with game-theoretic principles.

Reviewers valued the clear analysis, extensive experiments, and novel tool-integration for imperfect-information games. Key concerns centered on novelty, generalization, reward design, and evaluation robustness. The rebuttal has addressed most of the concerns. Overall, the strength of the paper overwhelms the weaknesses and the paper is recommended for acceptance.

**Reviewer Concerns:**

Key Concerns Raised:

- Novelty of ToolPoker (3gDV, whik, 4Bxf): How does it differ from prior tool-use frameworks (e.g., ReTool) beyond task adaptation?
- Generalization (3gDV, 4eQy): Can it extend to other imperfect-information games or multi-player settings?
- Reward design transparency (whik): Lack of ablation/sensitivity analysis for composite reward components.
- Evaluation robustness (4Bxf): Over-reliance on synthetic data; untested on human-style/noisy inputs.
- LLM-as-a-Judge validation (4Bxf): Need for calibration, inter-rater agreement, and human alignment.
- ToolPoker vs. CFR performance (4eQy): Why does it not outperform CFR, and can tool-calling errors be mitigated?

Addressed in Rebuttal:

- Clarified ToolPoker’s uniqueness (unified solver API, low-cost expert dataset) and empirically outperformed ReTool.
- Demonstrated scalability to three-player Limit Texas Hold’em and outlined extension to Mahjong.
- Added reward component ablations and trajectory visualizations, showing Ranswer​ is the key driver.
- Clarified training/evaluation includes noisy/off-equilibrium data; explained inherent robustness to human inputs.
- Provided calibration details, inter-rater agreement (GPT-4.1-mini vs. GPT-5), and human-aligned validation.
- Identified error patterns (state mis-specification, action-solver misalignment) and proposed mitigation strategies.

Outstanding/Partially Addressed:

- Novelty debate persists (conceptual overlap with "LLM + tools" paradigms).
- No direct evaluation on real human gameplay (acknowledged as future work).
- Limited theoretical grounding for equilibrium convergence (authors clarify reliance on solver guarantees).

**Reviewer Scores:**

3gDV: 8 → Remains 8

whik: 4 → Likely  4-6

4Bxf: 4 → Likely 4–6

4eQy: 6 → Likely 6-8

---

### Decision · Program_Chairs · 2026-01-26

Accept (Poster)